# Evidence that a positive feedback loop drives centrosome maturation in fly embryos

Ines Alvarez-Rodrigo, Thomas L Steinacker, Saroj Saurya, Paul T Conduit[†], Janina Baumbach[‡], Zsofia A Novak, Mustafa G Aydogan, Alan Wainman, Jordan W Raff*

The Sir William Dunn School of Pathology, University of Oxford, Oxford, United Kingdom

**Abstract** Centrosomes are formed when mother centrioles recruit pericentriolar material (PCM) around themselves. The PCM expands dramatically as cells prepare to enter mitosis (a process termed centrosome maturation), but it is unclear how this expansion is achieved. In flies, Spd-2 and Cnn are thought to form a scaffold around the mother centriole that recruits other components of the mitotic PCM, and the Polo-dependent phosphorylation of Cnn at the centrosome is crucial for scaffold assembly. Here, we show that, like Cnn, Spd-2 is specifically phosphorylated at centrosomes. This phosphorylation appears to create multiple phosphorylated S-S/T(p) motifs that allow Spd-2 to recruit Polo to the expanding scaffold. If the ability of Spd-2 to recruit Polo is impaired, the scaffold is initially assembled around the mother centriole, but it cannot expand outwards, and centrosome maturation fails. Our findings suggest that interactions between Spd-2, Polo and Cnn form a positive feedback loop that drives the dramatic expansion of the mitotic PCM in fly embryos.

DOI: https://doi.org/10.7554/eLife.50130.001

*For correspondence:
jordan.raff@path.ox.ac.uk

Present address: †Department of Zoology, University of Cambridge, Cambridge, United Kingdom; ‡Structural and Computational Biology Unit, European Molecular Biology Laboratory, Heidelberg, Germany

Competing interests: The authors declare that no competing interests exist.

## Introduction

Centrosomes play an important part in many aspects of cell organisation, and they form when a mother centriole recruits pericentriolar material (PCM) around itself (*Conduit et al., 2015*). The PCM contains several hundred proteins (*Alves-Cruzeiro et al., 2014*), allowing the centrosome to function as a major microtubule (MT) organising centre, and also as an important coordination centre and signalling hub (*Chavali et al., 2014*; *Vertii et al., 2016*). Centrosome dysfunction has been linked to several human diseases and developmental disorders, including cancer, microcephaly and dwarfism (*Bettencourt-Dias et al., 2011*; *Nigg and Raff, 2009*).

During interphase, the mother centriole recruits a small amount of PCM that is highly organised (*Lawo et al., 2012*; *Mennella et al., 2012*; *Sonnen et al., 2012*; *Fu and Glover, 2012*). As cells prepare to enter mitosis, however, the PCM expands dramatically around the mother centriole in a process termed centrosome maturation (*Palazzo et al., 2000*). Electron microscopy (EM) studies suggest that centrioles organise an extensive 'scaffold' structure during mitosis that surrounds the mother centriole and recruits other PCM components such as the γ-tubulin ring complex (γ-TuRC) (*Moritz et al., 1995*; *Moritz et al., 1998*; *Schnackenberg et al., 1998*).

In the fruit fly *Drosophila melanogaster* and the nematode *Caenorhabditis elegans*, a relatively simple pathway seems to govern the assembly of this mitotic PCM scaffold. The conserved centriole/centrosome protein Spd-2/SPD-2 (fly/worm nomenclature) cooperates with a large, predominantly predicted-coiled-coil, protein (Cnn in flies, SPD-5 in worms) to form a scaffold whose assembly is stimulated by the phosphorylation of Cnn/SPD-5 by the mitotic protein kinase Polo/PLK-

1 (*Conduit et al., 2014a*; *Conduit et al., 2014b*; *Feng et al., 2017*; *Woodruff et al., 2017*; *Woodruff et al., 2015*; *Wueseke et al., 2016*). Mitotic centrosome maturation is abolished in the absence of this pathway, and some aspects of Cnn and SPD-5 scaffold assembly have recently been reconstituted in vitro (*Feng et al., 2017*; *Woodruff et al., 2015*; *Woodruff et al., 2017*). Vertebrate homologues of Spd-2 (Cep192) (*Gomez-Ferreria et al., 2007*; *Zhu et al., 2008*), Cnn (Cdk5Rap2/Cep215) (*Barr et al., 2010*; *Choi et al., 2010*; *Fong et al., 2008*; *Kim and Rhee, 2014*; *Lizarraga et al., 2010*) and Polo (Plk1) (*Haren et al., 2009*; *Lane and Nigg, 1996*; *Lee and Rhee, 2011*) also have important roles in mitotic centrosome assembly, indicating that elements of this pathway are likely to be conserved in higher metazoans. In vertebrate cells another centriole and PCM protein, Pericentrin, also has an important role in mitotic centrosome assembly that is dependent upon its phosphorylation by Plk1 (*Haren et al., 2009*; *Lee and Rhee, 2011*). Pericentrin can interact with Cep215/Cnn (*Buchman et al., 2010*; *Kim and Rhee, 2014*; *Lerit et al., 2015*), but in flies the Pericentrin-like-protein (Plp) has a clear, but relatively minor, role in mitotic PCM assembly when compared to Spd-2 and Cnn (*Lerit et al., 2015*; *Martinez-Campos et al., 2004*; *Richens et al., 2015*).

Although most of the main players in mitotic centrosome-scaffold assembly appear to have been identified, several fundamental aspects of the assembly process remain mysterious. Cells entering mitosis, for example, contain two mother centrioles that assemble two mitotic centrosomes of equal size. It is unclear how this is achieved, as even a slight difference in the initial size of the two growing centrosomes would be expected to lead to asymmetric centrosome growth—as the larger centrosome would more efficiently compete for scaffolding subunits (*Conduit et al., 2015*; *Zwicker et al., 2014*). The centrioles in fly embryos appear to overcome this problem by constructing the PCM scaffold from the 'inside-out': Spd-2 and Cnn are only incorporated into the scaffold close to the mother centriole, and they then flux outwards to form an expanded scaffold around the mother centriole (*Conduit et al., 2014b*; *Conduit et al., 2010*). In this way, the growing PCM scaffold could ultimately attain a consistent steady-state size—where incorporation around the mother centriole is balanced by loss of the scaffold at the centrosome periphery—irrespective of any initial size difference in the PCM prior to mitosis (*Conduit et al., 2015*; *Raff, 2019*).

A potential problem with this 'inside-out' mode of assembly is that the rate of centrosome growth is limited by the very small size of the centriole. Mathematical modelling indicates that the incorporation of a crucial PCM scaffolding component only around the mother centriole cannot easily account for the high rates of mitotic centrosome growth observed experimentally (*Zwicker et al., 2014*). To overcome this problem, it has been proposed that centrosome growth is 'autocatalytic', with the centriole initially recruiting a key scaffolding component that can subsequently promote its own recruitment (*Woodruff et al., 2014*; *Zwicker et al., 2014*). It has been proposed that Spd-2 and Cnn could form a positive feedback loop that might serve such an autocatalytic function: Spd-2 helps recruit Cnn into the scaffold, and Cnn then helps to maintain Spd-2 within the scaffold, thus allowing higher levels of Spd-2 to accumulate around the mother centriole, which in turn drives higher rates of Cnn incorporation (*Conduit et al., 2014b*; *Conduit et al., 2015*; *Raff, 2019*).

In worms (*Decker et al., 2011*) and vertebrates (*Joukov et al., 2010*; *Joukov et al., 2014*; *Meng et al., 2015*), SPD-2/Cep192 can help recruit PLK1/Plk1 to centrosomes and Cep192 also activates Plk1 in vertebrates, in part through recruiting and activating Aurora A, another mitotic protein kinase implicated in centrosome maturation. We suspected, therefore, that in flies Spd-2 might recruit Polo into the centrosome-scaffold to phosphorylate Cnn and so help to generate a positive feedback loop that drives the expansion of the mitotic PCM. In flies, however, no interaction between Polo and Spd-2 has been reported. Indeed, an extensive Y2H screen for interactions between key centriole and centrosome proteins identified interactions between Spd-2 and the mitotic kinases Aurora A and Nek2, and between Polo and the centriole proteins Sas-4, Ana1 and Ana2, but not between Polo and Spd-2 (*Galletta et al., 2016*). A possible explanation for this result is that Polo/Plk1 is believed to be largely recruited to its many different locations in the cell, including centrosomes, through its Polo-Box-Domain (PBD), which binds to phosphorylated S-S/T(p) motifs (*Elia et al., 2003a*; *Elia et al., 2003b*; *Lee et al., 1998*; *Liu et al., 2004*; *Reynolds and Ohkura, 2003*; *Seong et al., 2002*; *Song et al., 2000*). Perhaps any such Polo binding sites in fly Spd-2 were simply not phosphorylated in the Y2H experiments. In support of this possibility, phosphorylated S-S/T(p) motifs in SPD-2/Cep192 have previously been shown to help recruit PLK1/Plk1 to centrosomes in worms (*Decker et al., 2011*), frogs (*Joukov et al., 2010*) and humans (*Meng et al., 2015*).

Here, we examine the potential role of Spd-2 in recruiting Polo to centrosomes in *Drosophila* embryos. We find that, like Cnn, Spd-2 is largely unphosphorylated in the cytosol, but is highly phosphorylated at centrosomes, where Spd-2 and Polo extensively co-localise within the pericentriolar scaffold. We show that a Spd-2 fragment containing 19 S-S/T motifs exhibits enhanced binding to the PBD in vitro when it has been phosphorylated by Plk1, but no enhancement is seen if these S-S/T motifs are mutated to T-S/T—a mutation that strongly perturbs PBD binding (*Elia et al., 2003b*). We express forms of Spd-2 in vivo in which either all 34 S-S/T motifs, or the 16 most conserved S-S/T motifs, have been mutated to T-S/T to perturb PBD-binding. These mutant Spd-2 proteins are still recruited to mother centrioles, as are Polo and Cnn, and these proteins assemble a PCM scaffold around the mother centriole. Strikingly, however, this PCM scaffold can no longer expand outwards, and centrosome maturation fails. These observations provide strong support for the hypothesis that Spd-2, Polo and Cnn cooperate to form a positive feedback loop that is required to drive the rapid expansion of the mitotic PCM in fly embryos.

## Results

### Spd-2 is phosphorylated specifically at centrosomes

We showed previously that Cnn is specifically phosphorylated at centrosomes (*Conduit et al., 2014a*), so we wondered if this was also the case for Spd-2. We partially purified centrosomes from embryo extracts by sucrose step-gradient centrifugation and compared the electrophoretic mobility of Spd-2 on western blots of the gradient fractions (*Figure 1A*). As was the case for Cnn, we observed a prominent slower migrating form of Spd-2 in the heavier centrosomal fractions that was largely absent in the lighter cytosolic fractions. However, unlike Cnn, a faster migrating form of Spd-2 was also present in the centrosomal fractions. Treatment of the centrosomal fractions with phosphatase revealed that the reduced mobility of Spd-2 in the centrosomal fractions could be attributed to phosphorylation (*Figure 1B*). Thus, like Cnn, Spd-2 is specifically phosphorylated at centrosomes, although not all the Spd-2 at the centrosome appears to be phosphorylated.

### Mutating multiple centrosomal phosphorylation sites in Spd-2 only mildly perturbs Spd-2 function in vivo

Using Mass Spectroscopy, we identified seven Spd-2 peptides that were phosphorylated consistently and with high confidence in the centrosomal fractions, but not in the cytosolic fractions (*Table 1*). One of these peptides contained a phosphorylated S-S(p) motif which could potentially help recruit Polo to centrosomes via its PBD. We generated transgenic lines expressing WT Spd-2-GFP and a mutant form of Spd-2-GFP in which all seven of the centrosomally phosphorylated residues—together with an additional 4 Ser/Thr resides that could potentially be phosphorylated in these peptides (*Table 1*)—were mutated to Ala (**Spd-2-11A-GFP**). Interestingly, although Spd-2-11A-GFP was expressed at substantially lower levels than WT Spd-2-GFP (*Figure 1—figure supplement 1A*), both Spd-2-GFP and Spd-2-11A-GFP rescued the female sterility phenotype of *Spd-2* mutant flies, and in *Spd-2* mutant embryos the mutant protein localised to centrosomes nearly as well as the WT protein (*Figure 1—figure supplement 1B–D*). Thus, although present at lower levels, Spd-2-11A-GFP is recruited to centrosomes nearly as well as the WT protein and preventing the phosphorylation of at least seven sites in Spd-2 that are specifically phosphorylated at centrosomes appears to only mildly perturb Spd-2 function in vivo.

### Generating a mutant form of Spd-2 with reduced binding to the Polo PBD

The Spd-2 phosphorylation sites we identified here were also identified previously in at least one of several phospho-proteomic screens in *D. melanogaster* (*Bodenmiller et al., 2007*; *Habermann et al., 2012*; *Hu et al., 2019*; *Zhai et al., 2008*). These screens, however, also identified many additional phosphorylated peptides in Spd-2, including 16 peptides that contained at least one phosphorylated S-S/T(p) motif that could potentially bind the PBD (*Table 2*). We speculated, therefore, that *Drosophila* Spd-2 might utilise multiple phosphorylated S-S/T(p) motifs to help recruit Polo to centrosomes.

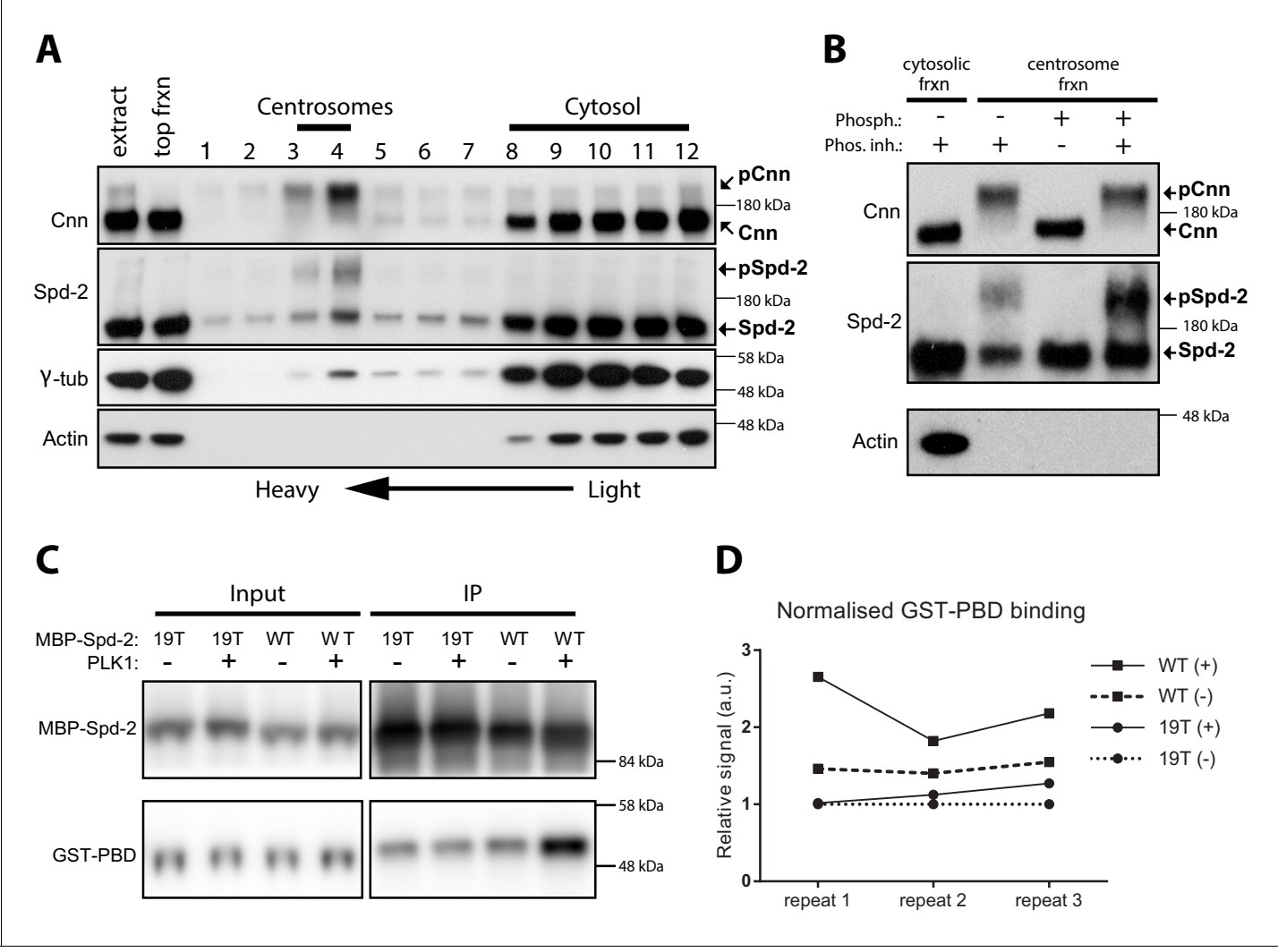

**Figure 1.** Spd-2 is phosphorylated at centrosomes and can bind the PBD in vitro in a manner that is enhanced by phoshorylation. (**A**) Western blot of a sucrose step-gradient purification of centrosomes from embryo extracts probed with anti-Cnn, Spd-2, γ-tubulin and Actin antibodies, as indicated. The gradient fractions are labelled 1 (heaviest) to 12 (lightest), and the cytosolic and centrosomal-peak fractions are indicated. (**B**) A western blot of centrosomal fractions from the step gradient treated with phosphatase (with or without phosphatase inhibitor), and probed with the indicated antibodies; the cytosolic fraction is also shown. The Cnn and Actin blots were presented previously (*Conduit et al., 2014a*) (reproduced here under a CC-BY 3.0 licence https://creativecommons.org/licences/by/3.0/), and were performed contemporaneously with the Spd-2 blot shown here. The blots shown are representative of two technical replicates for each of two biological repeats. (**C**) Western blot of an experiment in which recombinant WT MBP-Spd-2$_{(aa352-758)}$ or mutant MBP-Spd-2$_{(aa352-758)}$−19T were bound to MBP-Antibody-beads then either phosphorylated or not phosphorylated by human recombinant Plk1 (+/-) before mixing with human recombinant GST-PBD (Input). The beads were washed and any proteins still bound to the beads were eluted (IP). The Input and IP fractions were probed with either anti-Spd-2 antibodies (top panels) or anti-GST antibodies (bottom panels). (**D**) Graph shows the quantification of the amount of GST-PBD bound to the indicated beads (normalised to the amount of Spd-2 in each fraction) in three independent repeats; + /- indicates treatment with Plk1. To facilitate comparisons between repeats, the data is shown normalised to the signal from each 19T(-) sample. Enhanced binding of GST-PBD to the phosphorylated WT protein was observed in all three experiments, but was somewhat variable. Due to the variation and the small sample size (n = 3), we were unable to determine whether the data was normally distributed, so we could not apply parametric tests for statistical significance. Non-parametric tests did not indicate a significant increase in binding to the phosphorylated protein (p<0.05), but these tests do not work well with such a low number of repeats.
DOI: https://doi.org/10.7554/eLife.50130.002

The following source data and figure supplements are available for figure 1:

**Source data 1.** *Figure 1D* source data, quantification of GST-PBD signal.
DOI: https://doi.org/10.7554/eLife.50130.004

**Figure supplement 1.** Mutation of several centrosomal phosphosites in Spd-2 appears to have only a small effect on the centrosomal function of Spd-2 in vivo.

*Figure 1 continued on next page*

*Figure 1 continued*

DOI: https://doi.org/10.7554/eLife.50130.003

**Figure supplement 1—source data 1.** *Figure 1—figure supplement 1D* source data, quantification of centrosomal fluorescent intensity of either Spd-2-GFP or Spd-2-11A-GFP.

DOI: https://doi.org/10.7554/eLife.50130.005

As a first test of this hypothesis, we examined whether phosphorylated S-S/T motifs in Spd-2 could bind the PBD in vitro. Full-length Spd-2 fusion proteins were unstable when expressed in *E. coli*, so we purified an MBP-fusion containing approximately the middle 1/3 of Spd-2 (aa352-758, **MBP-Spd-2-WT**), a fusion that we had previously purified and raised antibodies against (*Dix and Raff, 2007*). This fusion protein contained 19 S-S/T motifs and, when phosphorylated by recombinant human Plk1 in vitro, it exhibited an enhanced ability to co-immunoprecipitate (IP) with recombinant GST-PBD (*Figure 1C*). Importantly, this phosphorylation-enhanced binding to GST-PBD was abolished when these 19 S-S/T motifs were mutated to T-S/T (**MBP-Spd-2–19T**)—a conservative substitution that has nevertheless been shown to perturb PBD-binding to these motifs (*Elia et al., 2003b*). Thus, a fragment of Spd-2 can bind directly to the PBD in a manner that is enhanced when the fragment is phosphorylated by Plk1, and this enhanced binding is prevented when the S-S/T motifs are mutated to T-S/T. Interestingly, this suggests that, in vitro at least, Plk1 can phosphorylate Spd-2 to 'prime' its own binding to Spd-2.

To test the potential role of Spd-2 in recruiting Polo to the mitotic PCM in vivo, we generated transgenic lines expressing a Spd-2-GFP fusion in which all 34 S-S/T motifs in *D. melanogaster* Spd-2 were mutated to T-S/T (**Spd-2-ALL-GFP**) (*blue* and *red* lines, **Figure 2A**). We reasoned that the conservative substitution of Thr for Ser might not disturb the overall folding of the protein, but that mutating all these sites should prevent them from binding the PBD efficiently (*Elia et al., 2003b*). In addition, 16 of the 34 S-S/T motifs were highly conserved in *Drosophila* species (*Figure 2A*, *red* lines), so we generated transgenic lines expressing a form of Spd-2-GFP in which only these 16 conserved motifs were mutated (**Spd-2-CONS-GFP**).

Western blotting revealed that the mutant Spd-2-GFP-fusions were expressed at slightly lower levels than WT Spd-2-GFP in embryos (*Figure 2B*). Importantly, however, both mutant fusion proteins rescued the defect in pronuclear fusion in *Spd-2* mutant embryos (although to a slightly lesser extent than the WT fusion protein), allowing these embryos to start to develop (*Figure 2C*). This demonstrates that the mutant proteins are not simply misfolded (see also below). Unlike Spd-2-11A-GFP, however, *Spd-2* mutant embryos expressing either Spd-2-CONS-GFP (hereafter **Spd-2-CONS-GFP embryos**) or Spd-2-ALL-GFP (hereafter **Spd-2-ALL-GFP embryos**) died early in embryonic development and almost never hatched as larvae (<1/500 embryos hatching; n > 2000 embryos scored).

**Table 1.** Identification of Spd-2 sites phosphorylated at the centrosome.

The Table lists amino acids in Spd-2 that were identified as being phosphorylated in the centrosomal fractions of embryo extracts, but not the cytosolic fractions, by Mass Spectroscopy. The Peptide score is the Mascot Ion Score (*Koenig et al., 2008*)—scores > 29 indicate identity or extensive homology (p<0.05). Phosphorylated amino acids are marked in *red*. These sites were mutated together with an additional 4 Ser/Thr resides that could potentially have been phosphorylated in these peptides (*blue*) to generate Spd-2-11A-GFP.

| Peptide sequence | Peptide score | High-scoring phosphorylated site | Additional sites mutated |
|---|---|---|---|
| GTNISFEPAEITGR | 53.03 | S121 | - |
| TNQPLLEPESNVTLDSVGEK | 65.26 | T329 | - |
| RPPSSSEILSLSAIDK | 38.85 | S397 | - |
| KPLSPLADHPQITISR | 34.55 | S484 | - |
| RVSIATMGLIPR | 29.93 | S569 | - |
| NLSPLSSPR | 42.33 | S614 | S617, S618 |
| GLGTSSVAVPR | 64.8 | S673 | T671, S672 |

DOI: https://doi.org/10.7554/eLife.50130.006

**Table 2.** The Table lists several previously identified Spd-2 peptides that include the potential PBD binding motif S-S/T(p). Definitively identified phosphorylated sites within PBD binding motifs are shown in red. Sites listed in brackets have been identified as phosphorylated but the scores were low (shown in blue); or have not been identified in *Drosophila melanogaster*, but have been definitively identified in other closely related species (shown in purple). Other phosphorylated sites which are on the same reported peptide, but are not part of a PBD binding motif, are shown in bold. The phospho-proteomic screens in which these peptides were identified are listed.

| Peptide sequence | Phospho sites | Other phospho sites | Ref. | Also |
|---|---|---|---|---|
| VFGDLSSFSKGRR | S34 S35 | (S37) | *Zhai et al., 2008* | - |
| ALETLEKPRPSRSSQAK | S76 | S73 | *Bodenmiller et al., 2007* | - |
| EKPSLSVAEILKSSFVEK | (S156) | S146 S148 | *Bodenmiller et al., 2007* | *Zhai et al., 2008* |
| SSSS | (S185) | | *Hu et al., 2019* | - |
| SENIWNIVSNSSPNRSR | S310 S311 | (S308) S315 | *Bodenmiller et al., 2007* | *Zhai et al., 2008; Hu et al., 2019* |
| RPPSSSEILSLSAIDK | (S389) (S390) S391 | (S395) S397 | *Bodenmiller et al., 2007* | *Zhai et al., 2008* |
| DIDLNSDTSTVEVVNHLWEHGR | (S413) (T414) | S410 (T412) | *Zhai et al., 2008* | - |
| ADTDPVETEAEADIDEWPSTPVKEPSRR | (S515) (T516) | T499 T504 S522 | | |
| AASPSSSDGVRPLTCTEDENDEEDEDKTPVNKK | S538 S539 (S540) | S536 (T547) (T549) T561 | | |
| KASSLSSTRLDGCDVAVASSTER | S581 S582 | (S584) | | |
| NLSPLSSPR | S617 S618 | S614 | *Bodenmiller et al., 2007* | *Zhai et al., 2008 Hu et al., 2019* |
| SCLSSPLLDSTTSSDRR | S624 S625 (S630) (T631) | | *Zhai et al., 2008* | *Hu et al., 2019* |
| SCLSSPLLDSTTSSDRR | S634 | | *Habermann et al., 2012* | - |
| ANSSPAGSEASSTSGFTASGR | S650 | S654 | *Bodenmiller et al., 2007* | - |
| KANSSPAGSEASSTSGFTASGR | (S658) | S654 | *Zhai et al., 2008* | - |
| RGLGTSSVAVPR | S673 S674 | (T672) | *Zhai et al., 2008* | This study; *Habermann et al., 2012* |

DOI: https://doi.org/10.7554/eLife.50130.007

## Spd-2-CONS-GFP and Spd-2-ALL-GFP embryos die early in development

To investigate why Spd-2-CONS-GFP and Spd-2-ALL-GFP embryos almost never hatched as larvae, we expressed a Jupiter-mCherry transgene in these embryos to follow the behaviour of centrosomes and MTs. Syncytial *Drosophila* embryos rapidly cycle between S- and M-phases without any Gap phases (*Foe and Alberts, 1983*). When the embryos exit mitosis they immediately enter S-phase and the centrosomes start to mature in preparation for the next round of mitosis; thus, the centrosomes in these syncytial embryos organise a relatively robust, mitotic-like, PCM at all stages. WT Spd-2-GFP localised strongly to centrosomes, and the centrosomes organised robust astral and spindle MT arrays (*Figure 3A*, left panels). The mutant proteins were less abundant at centrosomes (*Figure 3A,B*) and the astral MT arrays organised by the centrosomes were less robust (*Figure 3A, C*) (see 'Analysis of centrosome and MT fluorescence intensities' section in the Materials and methods for a full explanation of quantification methods); as a result, centrosomes were often detached

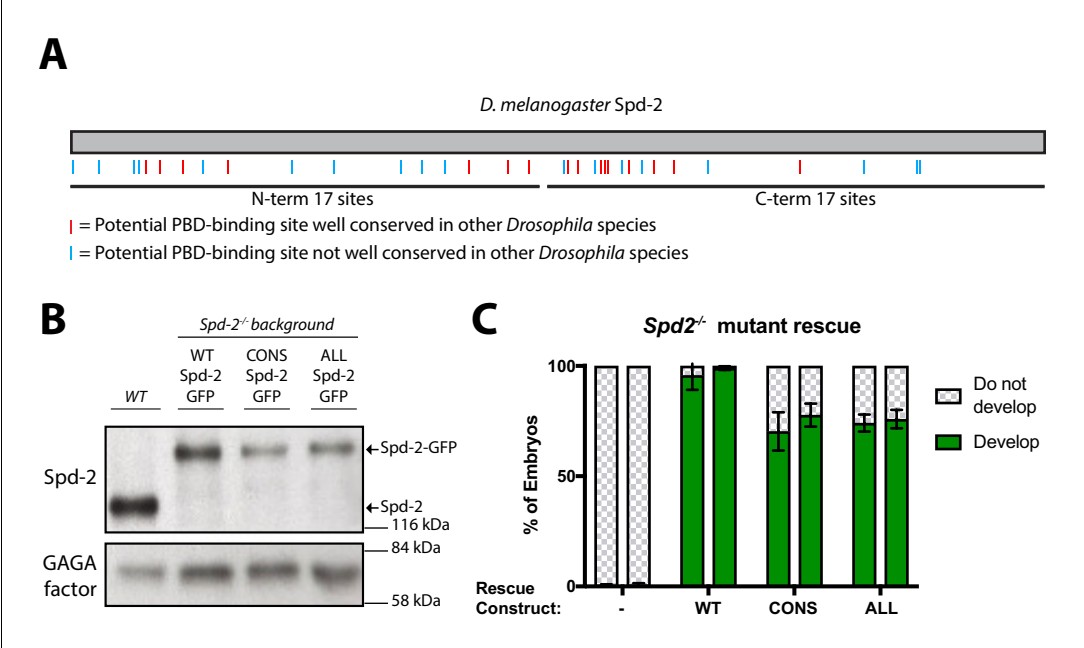

**Figure 2.** Generating forms of Spd-2 that should be unable to bind the PBD. (**A**) A schematic representation of *Drosophila melanogaster* Spd-2, indicating the position of S-S/T motifs that are either highly conserved (present in at least 11/12 *Drosophila* species analysed—*red* lines), or not highly conserved (*blue* lines). (**B**) Western blot of WT embryos, or *Spd-2* mutant embryos expressing either WT Spd-2-GFP, Spd-2-CONS-GFP or Spd-2-ALL-GFP, as indicated, probed with either anti-Spd-2 antibodies or anti-GAGA transcription factor antibodies (*Raff et al., 1994*) (as a loading control). The blot shown is representative of three technical replicates. (**C**) Bar charts quantify the percentage of *Spd-2* mutant embryos that had initiated development after expression of either WT Spd-2-GFP, Spd-2-CONS-GFP or Spd-2-ALL-GFP, as indicated. The chart shows the data from two independent biological repeats in which 3 lots of >50 embryos were collected and scored independently; error bars represent the standard deviation (SD).

DOI: https://doi.org/10.7554/eLife.50130.008

The following source data and figure supplements are available for figure 2:

**Source data 1.** *Figure 2A*, *Figure 2—figure supplement 1* source data; Spd-2 sequence alignment.
DOI: https://doi.org/10.7554/eLife.50130.010
**Source data 2.** *Figure 2C* source data, quantification of pronuclear arrest.
DOI: https://doi.org/10.7554/eLife.50130.015
**Figure supplement 1.** Distribution of S-S/T motifs in Spd-2/Cep192 and Cnn/Cep215 protein families.
DOI: https://doi.org/10.7554/eLife.50130.009
**Figure supplement 1—source data 1.** Cep192 (isoform 3) sequence alignment.
DOI: https://doi.org/10.7554/eLife.50130.011
**Figure supplement 1—source data 2.** Cep192 sequence alignment.
DOI: https://doi.org/10.7554/eLife.50130.012
**Figure supplement 1—source data 3.** Cep215 sequence alignment.
DOI: https://doi.org/10.7554/eLife.50130.013
**Figure supplement 1—source data 4.** Cnn sequence alignment.
DOI: https://doi.org/10.7554/eLife.50130.014

from the spindle poles (*arrowheads*, *Figure 3A*). Note that the reduced centrosomal levels of the Spd-2 mutant proteins are unlikely to be due to their lower expression levels (*Figure 2B*), as the Spd-2-GFP-11A mutant protein was also present at reduced levels, but localised to centrosomes nearly normally (*Figure 1—figure supplement 1*). Embryos expressing the mutant proteins exhibited progressively more severe mitotic defects as they developed, and these defects were qualitatively, but reproducibly, more pronounced in Spd-2-ALL-GFP embryos. We conclude that the mutant Spd-2-GFP fusions allow the assembly of a centrosome that can support pronuclear fusion, but these centrosomes cannot properly support the rapid syncytial divisions—and so the embryos accumulate mitotic defects and die during early development.

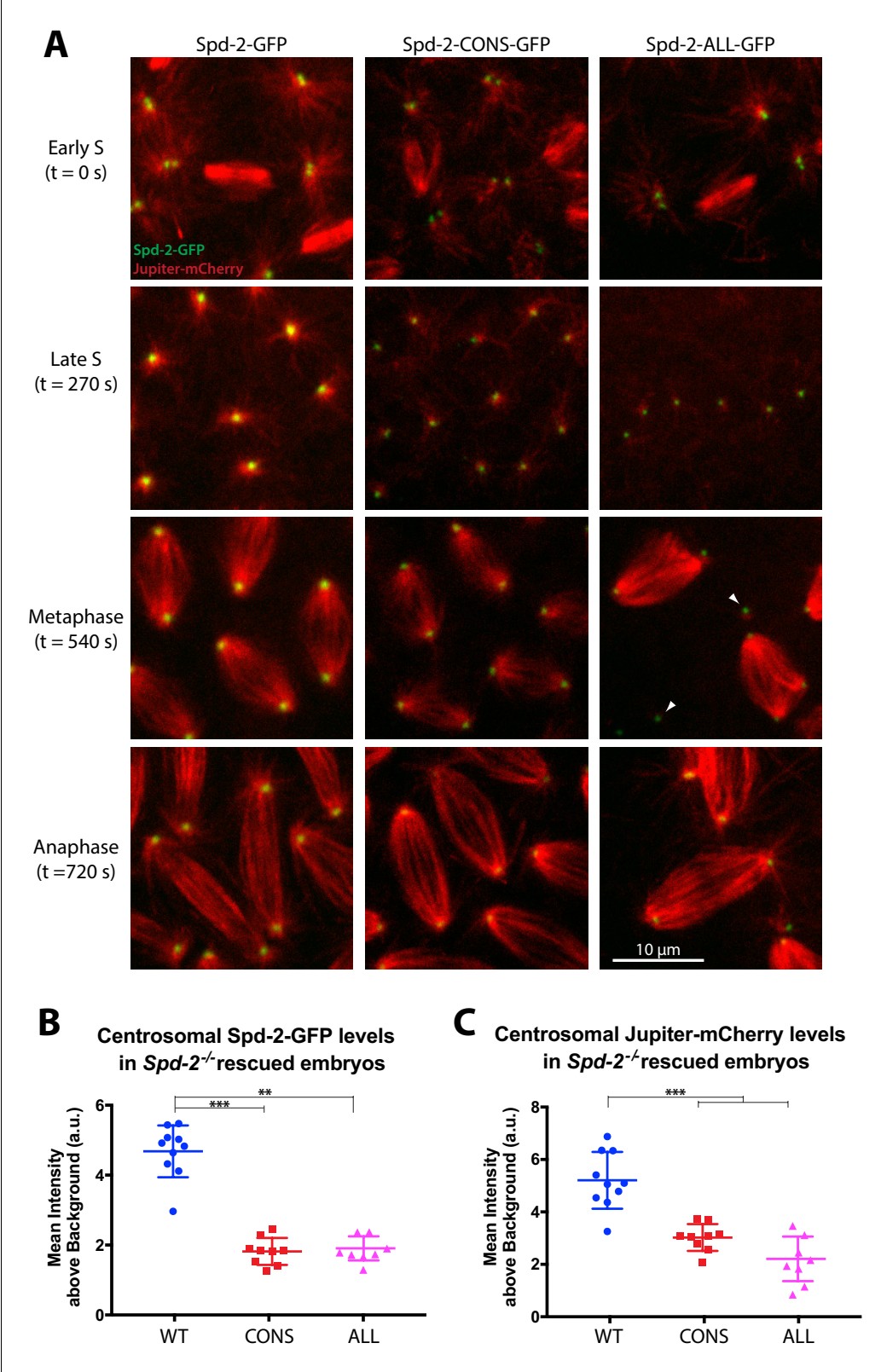

**Figure 3.** Centrosomes in Spd-2-CONS and Spd-2-ALL embryos recruit less Spd-2 and organise fewer MTs. (**A**) Micrographs show stills of living *Spd-2* mutant embryos expressing the MT-marker Jupiter-mCherry (*red*) and either WT Spd-2-GFP, Spd-2-CONS-GFP or Spd-2-ALL-GFP (*green*, as indicated). Time (in seconds) as the embryos progress from early S-phase (t = 0 s) to Anaphase (t = 720 s) is indicated. Embryos expressing the mutant proteins exhibited a range of mitotic defects, such as detached spindle poles (*white* arrowheads), that were more severe in Spd-2-ALL-GFP embryos. The Spd-2-
*Figure 3 continued on next page*

*Figure 3 continued*

CONS-GFP embryo shown here did not yet have any defects making it easier to compare to the WT-Spd-2-GFP embryo. (**B,C**) Graphs quantify the centrosomal intensity of the various Spd-2-GFP fusion proteins (**B**) or the centrosomal MT intensity (**C**) during late S-phase (t = 270 s). Each dot represents the average intensity of the five brightest centrosomes in a single embryo (n = 10, 9 and 8 embryos for WT, CONS and ALL embryos, respectively); error bars indicate the mean ± SD of each population of embryos scored. The D'Agostino–Pearson omnibus normality test was used to test for the Gaussian distribution of data. One-Way ANOVA with Tukey's multiple comparisons test was used when data passed the normality test (Jupiter-mCherry datasets); Kruskal-Wallis test with Dunn's multiple comparisons test was used otherwise (\*\*, $p < 0.01$; \*\*\*, $p < 0.001$).
DOI: https://doi.org/10.7554/eLife.50130.016
The following source data is available for figure 3:

**Source data 1.** *Figure 3B,C* source data; quantification of Spd-2-GFP and Jupiter-mCherry centrosomal intensity.
DOI: https://doi.org/10.7554/eLife.50130.017

## Spd-2-CONS-GFP and Spd-2-ALL-GFP are recruited to centrioles but cannot efficiently form a scaffold that spreads out from the centrioles

Spd-2 is localised to centrioles and to the mitotic PCM, so we wondered whether the reduction in the centrosomal levels of Spd-2-CONS-GFP and Spd-2-ALL-GFP was a result of a failure to properly localise these proteins to centrioles, to the PCM, or to both. Live-cell 3D-structured illumination super-resolution microscopy (3D-SIM) revealed that, as shown previously (*Conduit et al., 2014b*), WT Spd-2-GFP localised to the mother centriole, and also to a fibrous scaffold-like structure that extended outwards around the mother centriole (*Figure 4*, left panels). Strikingly, both mutant proteins still localised strongly to the mother centriole, but the extended scaffold-like structure appeared to be greatly reduced (*Figure 4*)—suggesting either that the mutant proteins were unable to efficiently incorporate into the scaffold, or that very little scaffold was assembled in these embryos.

To distinguish between these possibilities we tested whether the mutant Spd-2 proteins could co-assemble into a PCM scaffold formed by WT Spd-2. In *Spd-2* mutant embryos, we expressed one copy of WT *Spd-2-mCherry* and one copy of either *WT Spd-2-GFP*, *Spd-2-CONS-GFP* or *Spd-2-ALL-GFP*. Both mutant GFP-fusions co-assembled into a scaffold with WT Spd-2-mCherry in a manner that was very similar to the WT Spd-2-GFP (*Figure 5*). Thus, the mutant proteins can assemble into a Spd-2 scaffold if it is present, suggesting that the failure to detect a scaffold in Spd-2-CONS-GFP and Spd-2-ALL-GFP embryos (*Figure 4*) is likely to be due to the absence of the scaffold. Importantly, these observations also indicate that the mutant Spd-2 proteins are not misfolded, as they can clearly interact with the proteins that normally recruit and maintain Spd-2 at centrioles and within the mitotic PCM. A potential caveat to this interpretation is that Spd-2 proteins could form homo-oligomers, potentially allowing a largely misfolded mutant protein to oligomerise with a WT partner that then localises the mutant protein correctly. However, it has recently been shown using Fluorescence Correlation Spectroscopy (FCS) that SPD-2 is monomeric in worm embryos (*Wueseke et al., 2016*), and we found that this was also the case for Spd-2 in fly embryos (*Figure 5—figure supplement 1*). Thus, the mutant Spd-2 proteins are likely to be recruited to centrioles and centrosomes as monomers.

## A mitotic PCM scaffold is assembled in Spd-2-CONS-GFP and Spd-2-ALL-GFP embryos but it cannot expand efficiently around the centriole

We have previously hypothesised that Spd-2 cooperates with Polo and Cnn to form the mitotic PCM scaffold in flies (*Conduit et al., 2014b*; *Conduit et al., 2015*). We therefore tested whether Polo or Cnn could form a scaffold even when the Spd-2-ALL and Spd-2-CONS proteins cannot. We first examined the distribution of Polo-GFP in living *Spd-2* mutant embryos expressing either WT Spd-2-mCherry, Spd-2-CONS-mCherry or Spd-2-ALL-mCherry. The co-expression of Polo-GFP with the mutant Spd-2-mCherry proteins led to mitotic defects in embryos that were more severe than those observed when mutant GFP- or mCherry-fusions were expressed in the absence of Polo-GFP, suggesting that the GFP-tagged Polo sensitises the embryos to the expression of the mutant Spd-2-fusions (*Figure 6—figure supplement 1*). These defects were so severe in embryos expressing Spd-2-ALL-mCherry that we could not reliably stage them, so they were excluded from this analysis. WT Spd-2-mCherry and Polo-GFP extensively co-localised at the mother centriole and spread outwards

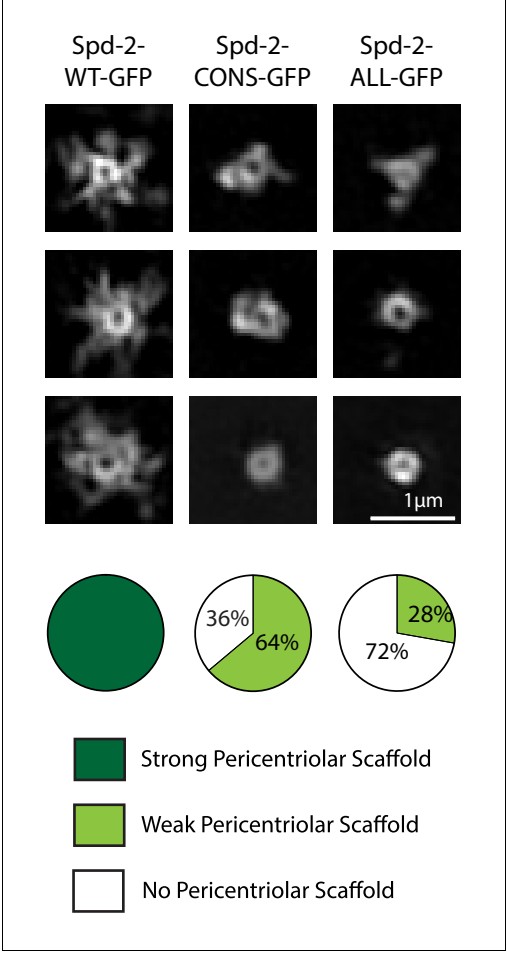

**Figure 4.** Spd-2-CONS and Spd-2-ALL do not efficiently assemble a PCM scaffold. Micrographs show 3D-SIM images of individual centrosomes from *Spd-2* mutant embryos expressing WT Spd-2-GFP, Spd-2-CONS-GFP or Spd-2-ALL-GFP (as indicated). Pie charts quantify the percentage of centrosomes that were scored qualitatively as having a strong (*dark green*), weak (*light green*) or no (*white*) pericentriolar scaffold (n = 36, for each genotype, respectively). In this, and all other SIM experiments, images were only included in the analysis if the reconstruction was deemed of sufficient quality by SIM-Check (*Ball et al., 2015*) (see Materials and methods for a full explanation of image quality control). All centrosomes were imaged in mid-late S-phase when the centrosomal levels of Spd-2 are maximal (see *Figure 8*). All scorings were performed blind by researchers not involved in the data acquisition.

DOI: https://doi.org/10.7554/eLife.50130.018

The following source data is available for figure 4:

**Source data 1.** *Figure 4* source data, results from blind scoring.

DOI: https://doi.org/10.7554/eLife.50130.019

together into the scaffold—supporting the idea that Spd-2 normally helps recruit Polo to the scaffold (*Figure 6A*). In contrast, although the Spd-2-CONS-mCherry and Polo-GFP proteins still co-localised around the mother centriole, neither protein formed a robust scaffold (*Figure 6B*). These observations suggest that phosphorylated S-S/T(p) motifs in Spd-2 are not required to recruit Polo to mother centrioles—and it is known that Polo can be recruited to centrioles by phosphorylated S-S/T motifs in at least one other centriole protein, Sas-4 (*Novak et al., 2016*)—but are required to recruit Polo to the PCM scaffold that expands around the mother centriole.

We next used 3D-SIM to examine the distribution of RFP-Cnn in living *Spd-2* mutant embryos expressing WT Spd-2-GFP or Spd-2-CONS-GFP. In WT Spd-2-GFP embryos, RFP-Cnn spread outwards along the centrosomal MTs forming a robust scaffold that extended beyond the Spd-2-GFP scaffold (*Figure 7A*), as reported previously (*Conduit et al., 2014b*). In contrast, in Spd-2-CONS-GFP embryos only an occasional protrusion of RFP-Cnn and Spd-2-CONS-GFP could be detected extending outwards from the centriole (*arrowheads*, *Figure 7B*). These relative distributions of RFP-Cnn were confirmed and quantified by 'radial-profiling' using data obtained on a standard spinning disk confocal system (*Conduit et al., 2014b*) (*Figure 7C*). Strikingly, radial-profiling also revealed how the RFP-Cnn scaffold (*red* lines, *Figure 7D*) normally extends beyond the Spd-2-GFP scaffold (*green* lines in *Figure 7D*) in WT embryos (*Figure 7Di*), but these distributions essentially overlap in Spd-2-CONS-GFP embryos (*Figure 7Dii*). Thus, if Spd-2 cannot efficiently recruit Polo via the PBD, the Cnn scaffold cannot efficiently expand outwards beyond the Spd-2 scaffold.

Cnn is normally phosphorylated at centrosomes in a Polo-dependent manner, and this allows Cnn to assemble into a scaffold (*Conduit et al., 2014a*; *Feng et al., 2017*). We tested whether Cnn could still be phosphorylated by Polo at centrosomes in Spd-2-CONS-GFP embryos using an antibody that specifically recognises a phospho-epitope in Cnn that is phosphorylated by Polo (*Feng et al., 2017*) (*Figure 7E,F*). Phosphorylated Cnn was still strongly detected around the mother centriole in Spd-2-CONS-GFP embryos. This phosphorylation is presumably dependent upon the Polo that is still recruited to the centrioles in Spd-2-CONS-GFP embryos (*Figure 6B*). We conclude that a Polo/Spd-2/Cnn 'mini-scaffold' assembles around the mother centriole even if Spd-2 cannot recruit

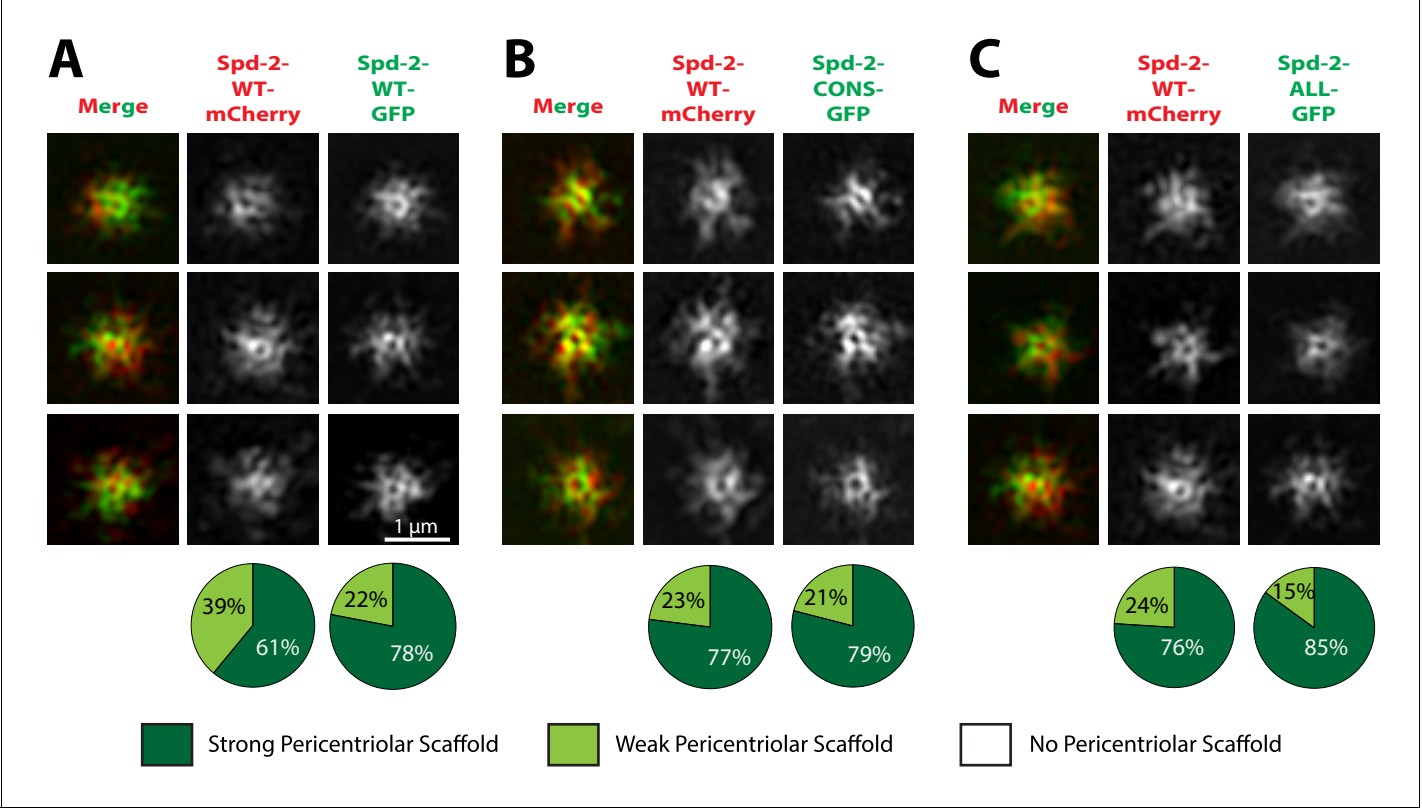

**Figure 5.** Spd-2-CONS and Spd-2-ALL can efficiently assemble into a PCM scaffold formed by WT Spd-2. (**A-C**) Micrographs show 3D-SIM images of individual centrosomes from *Spd-2* mutant embryos expressing WT Spd-2-mCherry and one copy of either WT Spd-2-GFP (**A**), Spd-2-CONS-GFP (**B**) or Spd-2-ALL-GFP (**C**). Pie charts quantify the percentage of centrosomes that were scored qualitatively as having a strong (*dark green*), weak (*light green*) or no (*white*) pericentriolar scaffold (n = 16 individual centrosomes, two images (channels) per centrosome, for each genotype, respectively). All centrosomes were imaged in mid-late S-phase when the centrosomal levels of Spd-2 are maximal (see *Figure 8*). All scorings were performed blind by researchers not involved in the data acquisition.

DOI: https://doi.org/10.7554/eLife.50130.020

The following source data and figure supplements are available for figure 5:

**Source data 1.** *Figure 5A–C* source data, results from blind scoring.
DOI: https://doi.org/10.7554/eLife.50130.022
**Figure supplement 1.** Spd-2-GFP diffuses as a monomer in the cytoplasm.
DOI: https://doi.org/10.7554/eLife.50130.021
**Figure supplement 1—source data 1.** *Figure 5—figure supplement 1* source data; FCS analysis.
DOI: https://doi.org/10.7554/eLife.50130.023

Polo via the PBD; this scaffold, however, is unable to efficiently expand outwards around the mother centriole.

As the Polo/Spd-2/Cnn scaffold is essential for mitotic centrosome assembly in flies (*Conduit et al., 2014b*; *Dobbelaere et al., 2008*; *Feng et al., 2017*), the inability of these proteins to form an expanded scaffold in Spd-2-CONS embryos should lead to a failure to recruit any other PCM proteins to an expanded mitotic PCM scaffold. This appeared to be the case, as the centrosomal recruitment of the PCM components Aurora A-GFP (*Figure 7—figure supplement 1A–D*) and γ-tubulin (*Figure 7—figure supplement 1E–H*) was greatly reduced in Spd-2-CONS-GFP embryos. Thus, the failure to form an expanded pericentriolar scaffold in these embryos appears to lead to a general failure in centrosome maturation.

## Centrosome maturation fails in Spd-2-CONS-GFP embryos

To directly examine the kinetics of centrosome maturation in Spd-2-CONS-GFP embryos we quantified the centrosomal levels of WT Spd-2-GFP and Spd-2-CONS-GFP through an entire embryonic

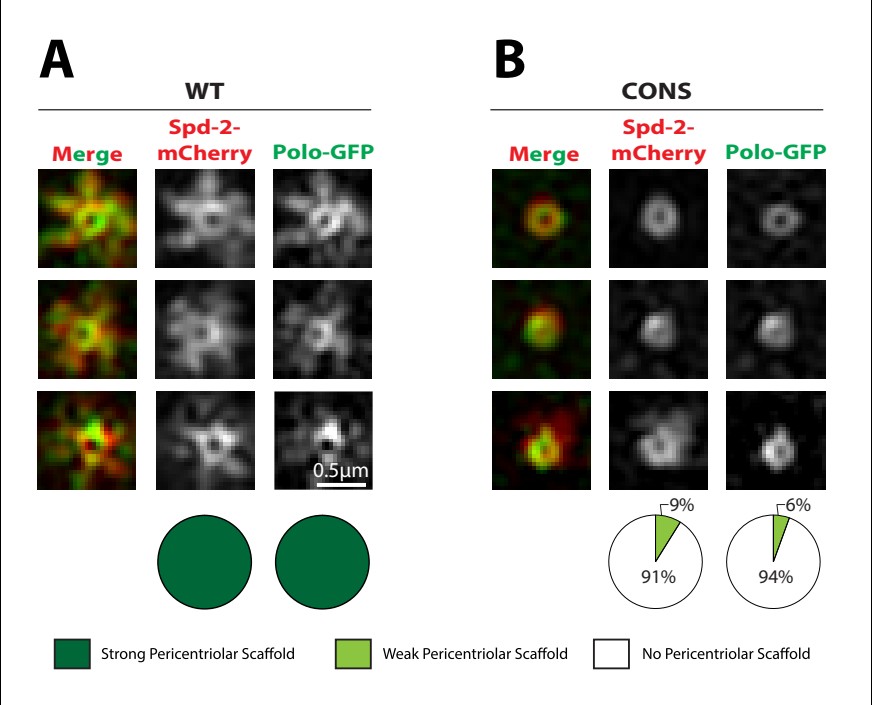

**Figure 6.** Polo is recruited to centrioles but cannot assemble into a PCM scaffold in Spd-2-CONS and Spd-2-ALL embryos. (**A,B**) Micrographs show 3D-SIM images of individual centrosomes from *Spd-2* mutant embryos expressing Polo-GFP (*green* in merged images) and either WT Spd-2-mCherry (**A**) or Spd-2-CONS-mCherry (**B**) (*red* in merged images). Pie charts quantify the percentage of centrosomes that were scored qualitatively as having a strong (*dark green*), weak (*light green*) or no (*white*) pericentriolar scaffold (n = 15 individual centrosomes, two images (channels) per centrosome, for each genotype, respectively). All centrosomes were imaged in mid-late S-phase when the centrosomal levels of Spd-2 are maximal (see *Figure 8*). All scorings were performed blind by researchers not involved in the data acquisition. The defect in scaffold assembly is stronger in the *Spd-2* mutant embryos expressing Polo-GFP and Spd-2-CONS-mCherry when compared to *Spd-2* mutant embryos expressing just Spd-2-CONS-GFP (*Figure 4*). This appears to be due to a genetic interaction between Polo-GFP and Spd-2-CONS-mCherry, as mutant embryos expressing just Spd-2-CONS-mCherry had a similar phenotype to embryos expressing just Spd-2-CONS-GFP (data not shown) (see main text).

DOI: https://doi.org/10.7554/eLife.50130.024

The following source data and figure supplement are available for figure 6:

**Source data 1.** *Figure 6* source data, results from blind scoring.
DOI: https://doi.org/10.7554/eLife.50130.026
**Figure supplement 1.** Abnormal Polo-GFP and Spd-2-mCherry distribution and mitotic defects in *Spd-2* mutant embryos rescued with Spd-2-CONS-mCherry or Spd-2-ALL-mCherry.
DOI: https://doi.org/10.7554/eLife.50130.025

cell-cycle. As described above, in these rapidly dividing syncytial embryos the two centrosomes separate at the start of S-phase and immediately start to mature in preparation for the next M-phase. WT Spd-2-GFP started to accumulate at the maturing centrosomes in early S-phase, reached maximal levels just before nuclear envelope breakdown, and then started to decline as the embryos entered mitosis (*blue* line, *Figure 8*; *Figure 8—figure supplement 1*). In contrast, centrosomal levels of Spd-2-CONS-GFP remained at a constant low level throughout the cycle (*red* line, *Figure 8*; *Figure 8—figure supplement 2*). These results suggest that centrosome maturation fails in Spd-2-CONS and Spd-2-ALL mutant embryos.

## Multiple regions of Spd-2 appear to help recruit Polo to the mitotic PCM scaffold

Studies in worms and frogs have concluded that a single S-S/T(p) motif in SPD-2/Cep192 is required to recruit PLK-1/Plk1 to centrosomes (*Decker et al., 2011*; *Joukov et al., 2010*; *Joukov et al.,*

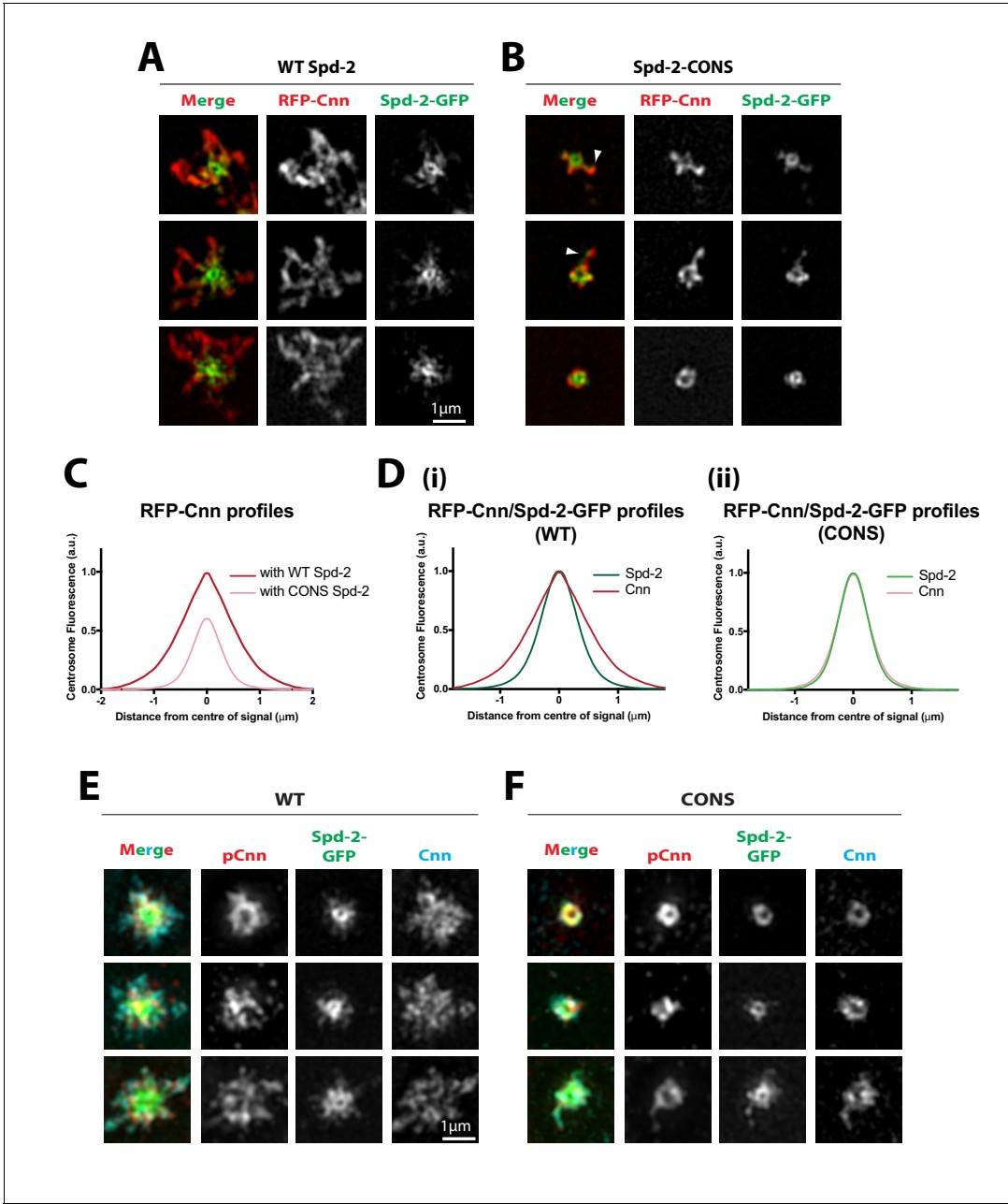

**Figure 7.** Cnn is recruited to, and phosphorylated at, centrioles in Spd-2-CONS embryos, but it does not efficiently assemble into an extended scaffold. (**A,B**) Micrographs show 3D-SIM images of individual centrosomes from living *Spd-2* mutant embryos expressing RFP-Cnn (*red* in merged images) and either WT Spd-2-GFP (A) or Spd-2-CONS-GFP (B) (*green* in merged images). Arrowheads indicate examples of occasional protrusions of RFP-Cnn and Spd-2-CONS-GFP. (**C,D**) Graphs compare the radial distributions of RFP-Cnn around the mother centriole in WT Spd-2-GFP and Spd-2-CONS-GFP embryos (C), or the radial distribution of RFP-Cnn and Spd-2-GFP in either WT Spd-2-GFP (D[i]) or Spd-2-CONS-GFP embryos (D[ii]). Data for these graphs was obtained from living embryos examined on a spinning disk confocal system; five centrosomes per embryo were analysed: n = 8 and 7 embryos for WT and CONS embryos, respectively. (**E,F**) Micrographs show 3D-SIM images of individual centrosomes from *Spd-2* mutant embryos expressing either WT Spd-2-GFP (E) or Spd-2-CONS-GFP (F) that were fixed and stained with antibodies against GFP, phospho-Cnn, or total-Cnn (*green*, *red* and *cyan* in merged images, respectively).

DOI: https://doi.org/10.7554/eLife.50130.027

The following source data and figure supplements are available for figure 7:

**Source data 1.** *Figure 7C,D* source data; radial profile analysis.

DOI: https://doi.org/10.7554/eLife.50130.029

**Figure supplement 1.** Evidence that the entire mitotic PCM does not expand outwards around the mother centriole in Spd-2-CONS embryos.

*Figure 7 continued on next page*

*Figure 7 continued*

DOI: https://doi.org/10.7554/eLife.50130.028

**Figure supplement 1—source data 1.** *Figure 7—figure supplement 1A,B* source data; radial profile analysis.

DOI: https://doi.org/10.7554/eLife.50130.030

**Figure supplement 1—source data 2.** *Figure 7—figure supplement 1E,F* source data; radial profile analysis.

DOI: https://doi.org/10.7554/eLife.50130.031

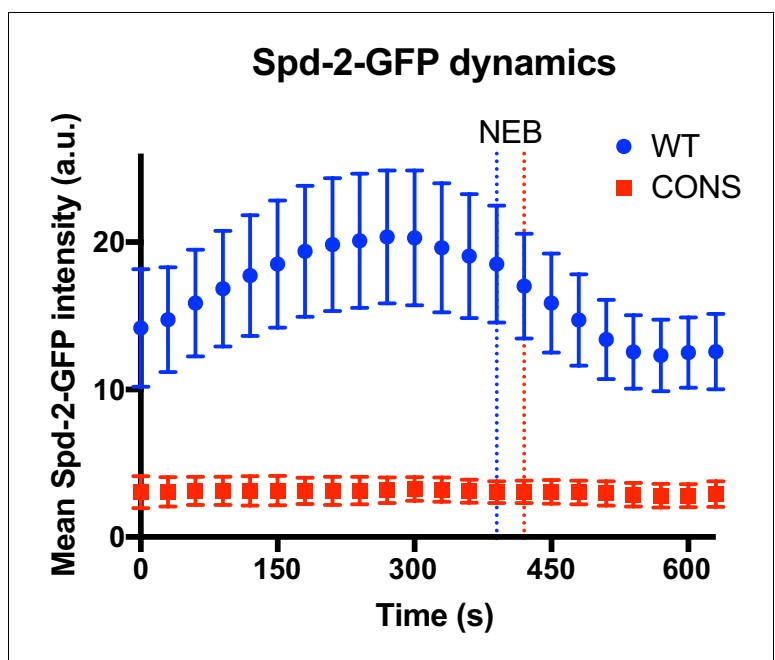

**Figure 8.** Centrosome maturation dynamics in Spd-2-CONS embryos. Graph compares the mean (± SD) centrosomal Spd-2-GFP intensity (in arbitrary units [a.u.]) through an entire embryonic cell cycle (nuclear cycle 12) in *Spd-2* mutant embryos expressing either WT Spd-2-GFP (*blue*) or Spd-2-CONS-GFP (*red*); n = 155 and 75 centrosomes analysed, respectively. Time in seconds is indicated, and the time when centrosomes first separate at the start of S-phase is set as t = 0; the time of mitotic entry—scored as the time of nuclear envelope breakdown (NEB)—is indicated by the dotted vertical lines. Because the length of S-phase varies in individual embryos—384s ± 46s or 369s ± 30s (mean ± SD) for WT Spd-2-GFP and Spd-2-GFP-CONS, respectively—we cannot simply average the data at each time point from multiple embryos, so representative embryos are shown here. The analysis of all 14 WT Spd-2-GFP (*blue*) or Spd-2-CONS-GFP (*red*) embryos that were monitored in this way is shown in *Figure 8—figure supplement 1* and *Figure 8—figure supplement 2*, respectively.

DOI: https://doi.org/10.7554/eLife.50130.032

The following source data and figure supplements are available for figure 8:

**Source data 1.** *Figure 8* source data, centrosomal Spd-2-GFP intensity through an entire embryonic cell cycle (embryos WT #13 and CONS #2).

DOI: https://doi.org/10.7554/eLife.50130.036

**Source data 2.** *Figure 8*, *Figure 8—figure supplement 1*, *Figure 8—figure supplement 2* source data; centrosomal Spd-2-GFP intensity through an entire embryonic cell cycle (all the embryos).

DOI: https://doi.org/10.7554/eLife.50130.037

**Figure supplement 1.** Centrosome maturation dynamics in *Spd-2* mutant embryos rescued by WT Spd-2-GFP.

DOI: https://doi.org/10.7554/eLife.50130.033

**Figure supplement 2.** Centrosome maturation dynamics in *Spd-2* mutant embryos rescued by Spd-2-GFP-CONS.

DOI: https://doi.org/10.7554/eLife.50130.034

**Figure supplement 3.** A schematic illustration of the models tested for the regression analysis of Spd-2-GFP dynamics during centrosome maturation.

DOI: https://doi.org/10.7554/eLife.50130.035

*2014*) while in human cells a second S-S/T(p) motif also plays a part (*Meng et al., 2015*). Our data, however, raise the possibility that multiple S-S/T(p) motifs in *Drosophila* Spd-2 may help recruit Polo to the mitotic PCM. To examine whether Polo recruitment by fly Spd-2 could be linked to any of the previous S-S/T(p) motifs identified, we used a previously established assay in which mRNAs encoding mKate2-tagged Spd-2-fusion proteins are injected into embryos expressing Polo-GFP; the mRNAs are gradually translated so the fusion proteins eventually out-compete the endogenous (unlabelled) WT Spd-2 and their effect on Polo-GFP recruitment can be assessed (*Novak et al., 2016*) (*Figure 9A*).

We first confirmed that the centrosomal recruitment of Polo-GFP was severely compromised by the injection of mRNAs encoding Spd-2-CONS-mKate2 or Spd-2-ALL-mKate2 (*Figure 9B,C*), and then assessed the contribution of various S-S/T motifs to Polo recruitment. The single S-S(p) motif in SPD-2 that recruits PLK-1 to centrosomes in *C. elegans* is a potential CDK1 substrate that may be conserved in *Drosophila* (T516, *blue* box, *Figure 9—figure supplement 1*). We mutated this site, together with the only other conserved S-S/T motif that is a potential CDK1 substrate in *Drosophila* Spd-2 (S625, *yellow* box, *Figure 9—figure supplement 1*) to Ala. This construct (Spd-2-AA-mKate2) did not detectably perturb the centrosomal distribution of Polo-GFP, suggesting that Spd-2 can recruit Polo to the PCM without any requirement that it first be primed to do so by Cdk1 phosphory-lation (*Figure 9—figure supplement 2*). Interestingly, all the Plk1-recruiting S-S/T motifs identified in worms and vertebrates are restricted to the N-terminal half of the protein (*Figure 9—figure supplement 1*). We therefore independently mutated the 17 S-S/T motifs in the N-terminal and C-terminal regions of Spd-2 to T-S/T (Spd-2-NT-mKate2 and Spd-2-CT-mKate2, respectively) (*Figure 2A*). Both constructs led to a reduction in Polo-GFP levels at the centrosome (*Figure 9D,E*). Spd-2-NT-mKate2 had the biggest effect, but this was still mild compared to Spd-2-ALL-mKate2 (*Figure 9D, E*). We conclude that there is no single S-S/T(p) motif in Spd-2 that is essential to recruit Polo to the mitotic PCM, and motifs in both the N- and C-terminal regions can contribute to this process.

## Spd-2-ALL and Spd-2-CONS cannot efficiently recruit Polo to the PCM, even when a PCM scaffold is present

An important caveat in interpreting our data is that we cannot be certain that the mutants prevent efficient mitotic PCM expansion because they cannot recruit Polo efficiently. Perhaps these muta-tions prevent PCM expansion for some other reason, and so Polo cannot be recruited to the expanded mitotic PCM because it simply does not exist in these embryos. Our mRNA injection assay potentially allowed us to address this issue. We looked for embryos injected with mRNA encoding Spd-2-CONS-mKate2 where the loss of Polo-GFP from the centrosome was just becoming apparent. We then used 3D-SIM to compare the centrosomal recruitment of mutant Spd-2-mKate2 and Polo-GFP to similarly staged controls injected with WT Spd-2-mKate2 mRNA (*Figure 10*). We found that Spd-2-CONS-mKate2 could often still be detected in an expanded PCM even when the amount of Polo recruited to the scaffold was severely reduced. Thus, in these centrosomes at least, an expanded PCM is still present (presumably because some unlabelled endogenous Spd-2 is still pres-ent), but the recruitment of Polo is severely reduced (presumably because the Spd-2-CONS-mKate2 in the scaffold cannot recruit Polo efficiently). These data strongly support our conclusion that the mutant Spd-2 proteins cannot efficiently recruit Polo to the expanded PCM.

## Discussion

Centrosome maturation appears to be a near-universal feature of the metazoan cell cycle. Although many of the key proteins required for centrosome maturation have been identified, how these pro-teins drive this process is unclear. We have previously proposed that three proteins—Spd-2, Polo and Cnn—together form a scaffold that expands around the mother centriole to recruit other PCM components to the mitotic centrosome (*Conduit et al., 2014b*; *Conduit et al., 2015*). The data pre-sented here suggests that these three proteins cooperate to form a positive feedback loop that drives the dramatic expansion of the mitotic PCM scaffold in fly embryos (*Figure 11A*).

We propose the following model (*Figure 11B*). In interphase cells, Spd-2, Polo and Cnn are recruited around the surface of the mother centriole (*Fu and Glover, 2012*), but Polo is inactive and Spd-2 and Cnn are not phosphorylated—so no scaffold is assembled (*Figure 11Bi*). As cells prepare to enter mitosis (*Figure 11Bii*), centrosomal Spd-2 becomes phosphorylated. Our in vitro data

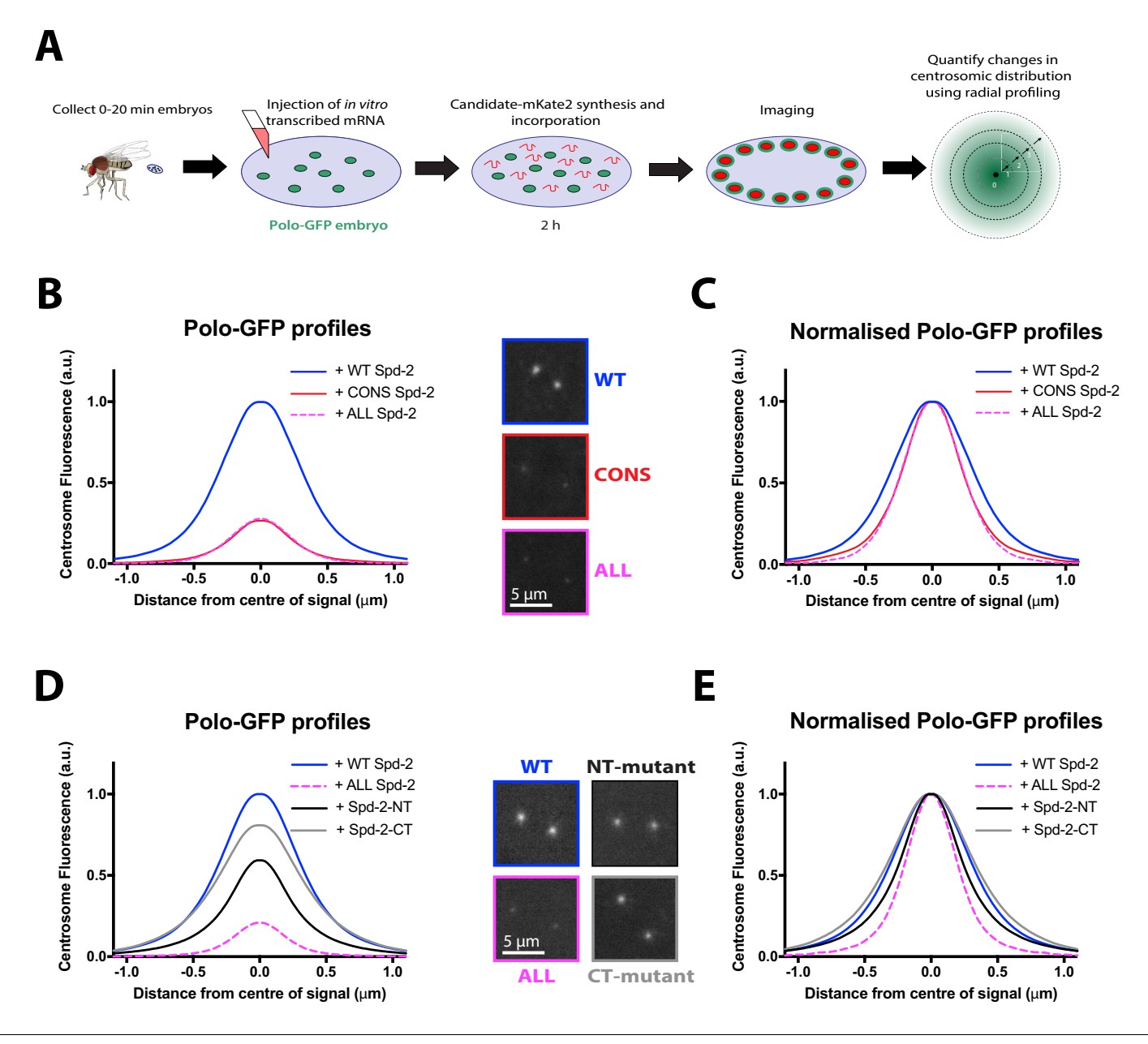

**Figure 9.** No single S-S/T motif in Spd-2 is essential for recruiting Polo to centrosomes. (**A**) Schematic illustration of the mRNA injection assay used to analyse the effect of various Spd-2-mKate2 fusion proteins on Polo-GFP recruitment. (**B**) Graph compares the radial distribution of Polo-GFP around the mother centriole in living WT embryos expressing Polo-GFP and injected with mRNAs encoding either WT Spd-2-mKate2, Spd-2-CONS-mKate2 or Spd-2-ALL-mKate2, as indicated; five centrosomes per embryo were analysed: n = 7, 6 and 9 embryos, respectively. Insets show examples of typical spinning disk confocal images used for this analysis. (**C**) Graph shows the same data as shown in (**B**), but normalised so that the peak intensity of all genotypes = 1. This emphasises how even if the centrosomal Polo-GFP signal is normalised for fluorescence intensity, Polo-GFP spreads out around the mother centriole to a lesser extent in the embryos expressing Spd-2-CONS-mKate2 or Spd-2-ALL-mKate2 than WT Spd-2-mKate2. These observations recapitulate our findings from transgenic lines expressing these Spd-2-GFP-fusions in a *Spd-2* mutant background. (**D,E**) Same analysis as presented in panels (B,C), but analysing the distribution of Polo-GFP in embryos expressing either Spd-2-ALL-mKate2, or one of two versions of Spd-2 in which the potential Polo binding sites in either the N-terminal or C-terminal region of Spd-2 have been mutated, as indicated; five centrosomes per embryo were analysed: n = 13, 6, 13 and 10 embryos, respectively.

DOI: https://doi.org/10.7554/eLife.50130.038

The following source data and figure supplements are available for figure 9:

**Source data 1.** *Figure 9B,C* source data; radial profie analysis (part 1).

*Figure 9 continued on next page*

*Figure 9 continued*

DOI: https://doi.org/10.7554/eLife.50130.041

**Source data 2.** *Figure 9B,C* source data; radial profie analysis (part 2).

DOI: https://doi.org/10.7554/eLife.50130.042

**Source data 3.** *Figure 9B,C* source data; radial profie analysis (part 3).

DOI: https://doi.org/10.7554/eLife.50130.043

**Source data 4.** *Figure 9B,C* source data; radial profie analysis (part 4).

DOI: https://doi.org/10.7554/eLife.50130.044

**Source data 5.** *Figure 9B,C* source data; radial profie analysis (part 5).

DOI: https://doi.org/10.7554/eLife.50130.045

**Source data 6.** *Figure 9D,E* source data, radial profile analysis.

DOI: https://doi.org/10.7554/eLife.50130.046

**Figure supplement 1.** A multiple sequence alignment (MSA) of Spd-2 protein sequences from different species highlighting the known PBD binding motifs.

DOI: https://doi.org/10.7554/eLife.50130.039

**Figure supplement 2.** Spd-2-AA-mKate2 does not detectably perturb the centrosomal distribution of Polo-GFP.

DOI: https://doi.org/10.7554/eLife.50130.040

**Figure supplement 2—source data 1.** Radial profile analysis.

DOI: https://doi.org/10.7554/eLife.50130.047

suggests that Polo is involved in this phosphorylation (via a 'self-priming and binding' mechanism), but other mitotic kinases may also be involved. Phosphorylation allows Spd-2 to form a scaffold (*red*, *Figure 11Bii*) that fluxes outwards (*red arrows*, *Figure 11Bii*) and that can recruit both Polo (via phosphorylated S-S/T(p) motifs) and Cnn (*Conduit et al., 2014b*). The active Polo (*blue dots*, *Figure 11Bii*) phosphorylates Cnn (orange arrow, *Figure 11A*), allowing it to also form a scaffold (*green*, *Figure 11Bii*) (*Conduit et al., 2014a*; *Feng et al., 2017*). The Spd-2 scaffold is inherently unstable (*Conduit et al., 2014b*), so it can only accumulate around the mother centriole if it is stabilised by the Cnn scaffold (*dotted arrow*, *Figure 11A*). The Cnn scaffold therefore allows the Spd-2 scaffold to expand outward, increasing Spd-2 levels within the PCM scaffold and allowing Spd-2 to recruit more Cnn and more Polo into the scaffold. This is a classical positive feedback loop in which the Output (the PCM scaffold *in toto*) directly increases the Input (the Spd-2 scaffold).

If Spd-2 cannot efficiently recruit Polo, as appears to be the case with the Spd-2-ALL and Spd-2-CONS mutants, it can still recruit Cnn, and this is, at least initially, phosphorylated by the pool of Polo that is still present around the mother centriole (*Figure 11C*). Our data suggests that this centriolar pool of Polo is not recruited by Spd-2 (at least not via the PBD), and we suspect that S-S/T(p) motifs in other centriole proteins, such as Sas-4 (*Novak et al., 2016*), normally recruit Polo to centrioles. As a result, mutant Spd-2 proteins can still support the assembly of a 'mini-scaffold' around the mother centriole, and this can recruit some PCM and organise some MTs (*Figure 11Cii*). The mutant Spd-2 scaffold that fluxes outwards from the mother centriole, however, cannot recruit Polo. Therefore the Cnn recruited by the expanding Spd-2 network cannot be phosphorylated, and it cannot form a scaffold to support the expanding Spd-2 network. As a result, the expanding mitotic PCM scaffold rapidly dissipates into the cytosol (*Figure 11Ciii*).

Although this mechanism is autocatalytic—as the expanding Spd-2 scaffold allows Polo and Cnn to be recruited into the PCM at an increasing rate—crucially, the mother centriole remains the only source of Spd-2 (*Figure 11A*). This potentially explains the conundrum of how mitotic PCM growth is autocatalytic (*Zwicker et al., 2014*), but at the same time requires the mother centriole (*Basto et al., 2006*; *Cabral et al., 2019*; *Kirkham et al., 2003*). This requirement for centrioles can also potentially explain how two spatially separated centrosomes usually grow their mitotic PCM to the same size (*Conduit et al., 2015*; *Raff, 2019*), as PCM size may ultimately be determined by how much Spd-2 can be provided by the centrioles, rather than how much PCM was present in the centrosome when maturation was initiated.

A key feature of this proposed mechanism is that Cnn cannot recruit itself or Spd-2 or Polo into the scaffold (although it helps to maintain the Spd-2 scaffold recruited by the centriole; *Figure 11A*). If it could do so, mitotic PCM growth would no longer be constrained by the centriole as Cnn could catalyse its own recruitment. Interestingly, although Spd-2 and Cnn are of similar size in flies (1146aa

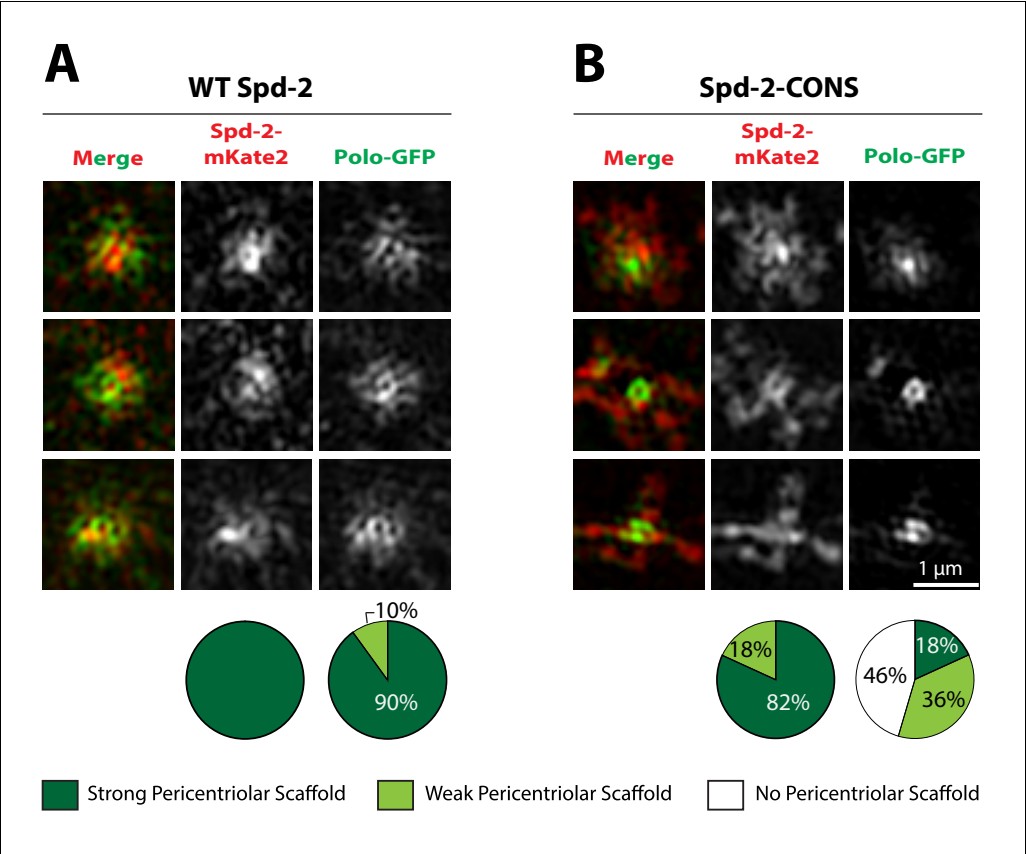

**Figure 10.** The recruitment of Polo-GFP to the PCM is perturbed in embryos expressing Spd-2-CONS-mKate2, even if a PCM scaffold is still detectable. (**A,B**) Micrographs show 3D-SIM images of individual centrosomes from embryos expressing Polo-GFP (*green* in merged images) injected with mRNA encoding either WT Spd-2-mKate2 (**A**) or Spd-2-CONS-mKate2 (**B**) (*red* in merged images). Pie charts quantify the percentage of centrosomes that were scored qualitatively as having a strong (*dark green*), weak (*light green*) or no (*white*) pericentriolar scaffold (n = 10 and 11 individual centrosomes, two images (channels) per centrosome, for WT and CONS injections, respectively). All centrosomes were imaged in mid-late S-phase when the centrosomal levels of Spd-2 are maximal (see *Figure 8*). All scorings were performed blind by researchers not involved in the data acquisition. Note that mKate2 is relatively slow folding, and the fusion proteins are just expressed from the injected mRNA, so the signal-to-noise ratio is low and the 3D-SIM images reconstruct relatively poorly. Nevertheless the presence of a Spd-2-CONS-mKate2 scaffold is clear, even when the Polo-GFP signal in the scaffold is very weak (**B**).
DOI: https://doi.org/10.7554/eLife.50130.048

The following source data is available for figure 10:

**Source data 1.** *Figure 10* source data, results from blind scoring.
DOI: https://doi.org/10.7554/eLife.50130.049

and 1148aa, respectively) Spd-2 has >5X more conserved potential PBD-binding S-S/T motifs than Cnn (*Figure 2—figure supplement 1*). Moreover, a similar ratio of conserved sites is found when comparing human Cep192 (1941aa) to human Cep215/Cdk5Rap2 (1893aa) (*Figure 2—figure supplement 1*), even though the human and fly homologues of both proteins share only limited amino acid identity. Perhaps, these two protein families have evolved to ensure that phosphorylated Spd-2/Cep192 can efficiently recruit Polo/Plk1, whereas phosphorylated Cnn/Cep215 cannot.

Our data indicates that multiple S-S/T(p) motifs in Spd-2 may be involved in Polo recruitment to the PCM. When only the most conserved motifs are mutated, other motifs in Spd-2 appear to be able to help recruit Polo, as evidenced by the additive effect of the Spd-2-ALL mutant compared to the Spd-2-CONS mutant. This mechanism of multi-site phosphorylation and recruitment could help amplify the maturation process (as the additional Polo recruited would allow Cnn to be phosphorylated at a higher rate) and so contribute to the establishment of the positive feedback loop.

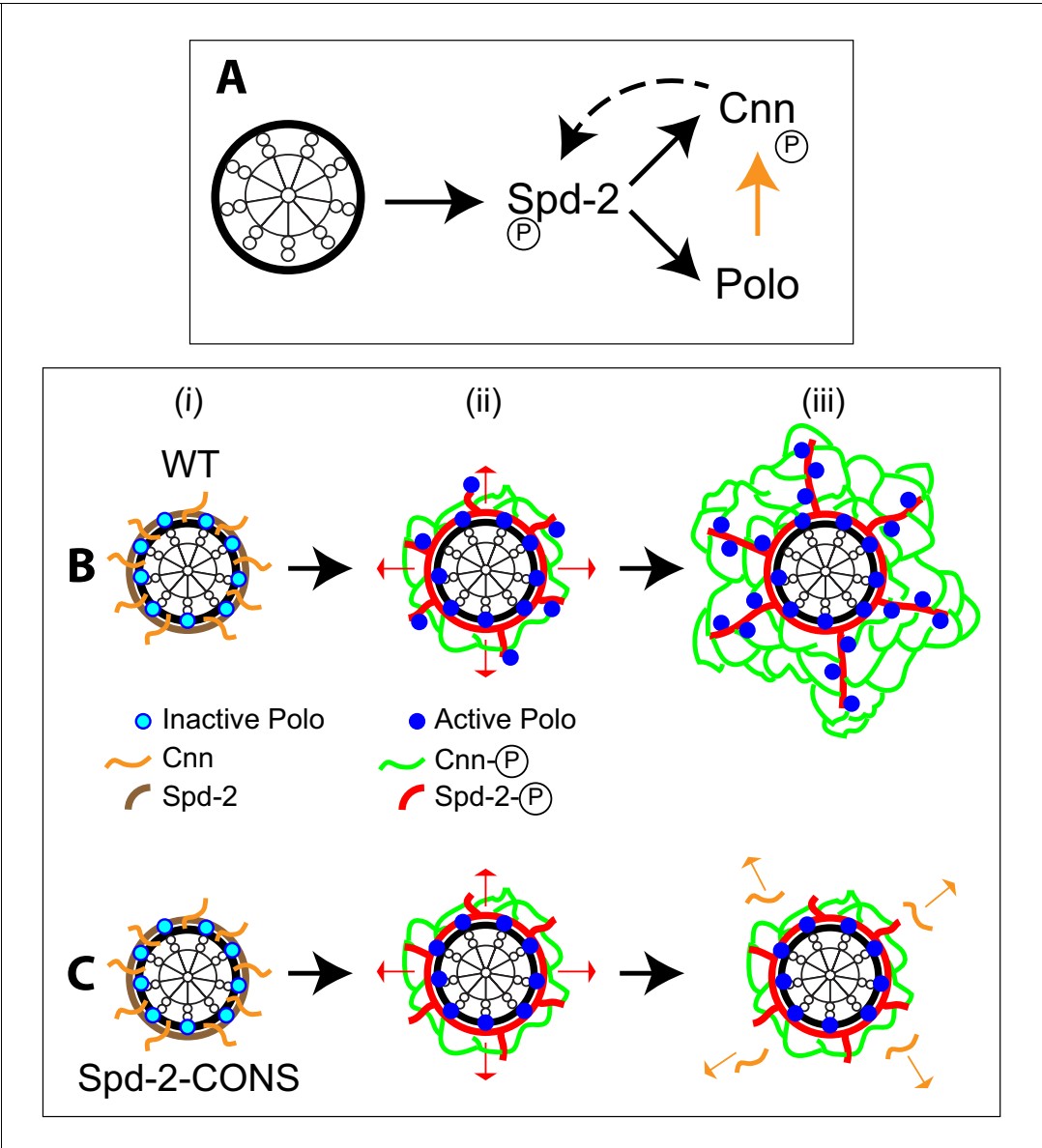

**Figure 11.** Spd-2, Polo and Cnn appear to form a positive feedback loop that drives the expansion of the mitotic PCM scaffold. (**A**) A schematic summary of the proposed positive feedback loop that drives the expansion of the mitotic PCM scaffold in *Drosophila* embryos. Solid black lines indicate recruitment, solid orange line indicates phosphorylation, dashed line indicates that Cnn does not *recruit* Spd-2, but rather helps to *stabilise* the Spd-2 scaffold that has been recruited to the mother centriole and is fluxing outwards. Polo may phosphorylate Spd-2 to create additional S-S/T(p) motifs that can then recruit more Polo, but this is not depicted here. This circuit is a classical positive feedback loop; ultimately, however, it relies on the mother centriole as a source of Spd-2, because Cnn itself cannot recruit more Spd-2 or Polo into the scaffold. (**B,C**) Schematics illustrate the process of centrosome maturation in a WT cell (B), and a cell in which Spd-2 cannot recruit Polo (C). During interphase (i), Spd-2, Polo and Cnn are all recruited to the mother centriole. Polo is inactive, Spd-2 and Cnn are not phosphorylated, so no scaffold forms. As cells prepare to enter mitosis (ii), Polo is activated, and the centrosomal Spd-2 and Cnn are phosphorylated, allowing them to initially assemble into a 'mini-scaffold' around the mother centriole. The phosphorylated Spd-2 scaffold then starts to flux away from the mother centriole (*red arrows*). In normal cells (B[iii]), the expanding Spd-2 scaffold recruits more Cnn and more Polo, allowing more Cnn scaffold to assemble. The Cnn scaffold cannot recruit more Spd-2 or Polo, but it stabilises the expanding Spd-2 scaffold; this allows the Spd-2 scaffold to accumulate around the mother centriole. This creates a positive feedback loop that drives an increasing rate of expansion of the Spd-2 and Cnn scaffolds around the mother centriole. If the expanding Spd-2 scaffold cannot recruit Polo (C[iii]), the Cnn recruited to the expanding Spd-2 scaffold is too far away from the centriole to get phosphorylated by Polo so it cannot form a scaffold and rapidly dissipates into the cytosol (*orange* arrows). The positive feedback loop is broken, and centrosome maturation fails.
DOI: https://doi.org/10.7554/eLife.50130.050

Another important feature of this proposed mechanism is that Spd-2 is incorporated into the mitotic PCM at the centriole surface and then fluxes outwards (*Conduit et al., 2014b*). This Spd-2-flux has so far only been observed in *Drosophila* embryos and mitotic brain cells (*Conduit et al., 2015*; *Conduit et al., 2010*; *Conduit et al., 2014b*). In fly embryos, Cnn also fluxes outwards but, unlike Spd-2, this flux requires MTs and is only observed in embryos (*Conduit and Raff, 2015*). In *C. elegans* embryos, SPD-5 behaves like Cnn in somatic cells: it does not flux outwards and is incorporated isotropically throughout the volume of the PCM (*Laos et al., 2015*). Moreover, a very recent study found no evidence for an outward centrosomal flux of SPD-2 in worm embryos (*Cabral et al., 2019*). Clearly, it will be important to determine whether Spd-2/Cep192 homologues flux outwards in other species and, if so, whether this flux provides the primary mechanism by which the mother centriole influences the growth of the expanding mitotic PCM.

In vertebrates, Cep192 serves as a scaffold for Plk1 and also Aurora A (*Joukov et al., 2010*; *Joukov et al., 2014*; *Meng et al., 2015*)—another mitotic protein kinase that plays an important part in centrosome maturation in many species (*Barr and Gergely, 2007*). There appears to be a complex interplay between Cep192, Plk-1 and Aurora A in vertebrates, with Cep192 acting as a scaffold that allows these two important regulators of mitosis to influence each other's activity and centrosomal localisation. Spd-2 clearly plays an important part in recruiting Aurora A to centrosomes in fly cells (*Conduit et al., 2014b*; *Dobbelaere et al., 2008*)—although it is unclear if this is direct, as fly and worm Spd-2/SPD-2 both lack the N-terminal region in vertebrate Cep192 that recruits Aurora A (*Meng et al., 2015*). How Aurora A might influence the assembly of the Spd-2, Polo/PLK-1 and Cnn/SPD-5 scaffold remains to be determined, although in worms AIR-1 (the Aurora A homologue) is required to initiate centrosome maturation, but is not required for subsequent PCM growth (*Cabral et al., 2019*).

Finally, there has been great interest recently in the idea that many non-membrane bound organelles like the centrosome may assemble as 'condensates' formed by liquid-liquid phase separation (*Banani et al., 2017*; *Boeynaems et al., 2018*; *Raff, 2019*). In support of this possibility for the centrosome, purified recombinant SPD-5 can assemble into condensates in vitro that have transient liquid-like properties, although they rapidly harden into a more viscous gel- or solid-like phase (*Woodruff et al., 2017*). Moreover, a mathematical model that describes centrosome maturation in the early worm embryo treats the centrosome as a liquid, and it is from this model that the importance of autocatalysis was first recognised (*Zwicker et al., 2014*). In vivo, however, the Cnn and SPD-5 scaffolds do not appear to be very liquid-like (*Conduit et al., 2010*; *Conduit et al., 2014b*; *Laos et al., 2015*) and fragments of Cnn can assemble into micron-scale assemblies in vitro that are clearly solid- or very viscous-gel-like (*Feng et al., 2017*). Our data suggests that the incorporation of Spd-2 into the PCM only at the surface of the centriole, coupled to an amplifying Spd-2/Polo/Cnn positive feedback loop, could provide an 'autocatalytic' mechanism that functions within the conceptual framework of a non-liquid-like scaffold that emanates from the mother centriole.

## Materials and methods

### Key resources table
See *Supplementary file 1*.

### Fly husbandry, stocks and handling
Flies were kept at 25˚C or 18˚C on *Drosophila* culture medium (0.77% agar, 6.9% maize, 0.8% soya, 1.4% yeast, 6.9% malt, 1.9% molasses, 0.5% propionic acid, 0.03% ortho-phosphoric acid and 0.3% nipagin). Stocks were kept in 8 cm x 2.5 cm plastic vials or 0.25-pint plastic bottles. Embryos were collected on cranberry-raspberry juice plates (25% cranberry-raspberry juice, 2% sucrose and 1.8% agar) supplemented with fresh yeast. Standard fly handling techniques were employed (*Roberts, 1998*). In vivo studies were performed using 1.5–2 hr-old syncytial blastoderm stage embryos. After 0–1 hr collections at 25˚C, embryos were aged at 25˚C for 30–60 min. When injecting mRNA, embryos were collected for 20 min, injected, and imaged after 120–150 min at 21˚C (but always within the syncytial blastoderm stage of development). Prior to injection or imaging, embryos were dechorionated on double-sided tape and mounted on a strip of glue onto a 35-mm glass-bottom petri dish with a 14 mm micro-well (MatTek). After desiccation for 1 min (non-injection experiments)

or 3 min (pre-mRNA injection) at 25°C, embryos were covered in Voltalef oil (ARKEMA). Live imaging was performed using either the spinning disk confocal or the 3D-SIM systems described below.

## Transgenic *Drosophila* lines

Potential Polo binding sites in the amino acid sequence of *Drosophila melanogaster* Spd-2 were identified by searching for the consensus Polo binding motif S-S/T. Site conservation was assessed using FlyBase BLAST (selecting the genus *Drosophila*) and Jalview for protein alignment. The ALL and CONS constructs were designed in silico and synthesised externally by GENEWIZ Co. Ltd. (Suzhou, China); the WT *spd-2* cDNA was obtained from Geneservice Ltd (UK). The WT, ALL and CONS Spd-2 cDNAs were cloned into a pDONR-Zeo vector and then introduced in Ubq-GFPCT and Ubq-mCherryCT destination vectors via Gateway cloning as indicated (Key Resources Table). The Ubq-Spd-2-11A-GFP plasmid was derived via site-directed mutagenesis on pDONRSpd-2 vector using QuikChange Multi Site-Directed mutagenesis followed by Gateway cloning into the Ubq-GFPCT vector. The plasmids for monomeric and dimeric NeonGreen expressed from the Sas-6 or Plk4 promoter were generated using the NEBuilder HiFi DNA Assembly (New England Biolabs). The fluorophores of dimeric NeonGreen were linked with a five amino acid-long peptide to minimise energy transfer between them. The 2 kb upstream of the start codon and 1 kb downstream of the stop codon of Sas-6/Plk4 were amplified from Oregon-R genomic DNA. The amplified fragments were cloned into the pDONR-Zeo vector. The plasmids were sent to BestGene Inc (Chino Hills, California) or the University of Cambridge Genetics Fly Facility (UK) for generation of the transgenic lines via random P-element insertion in to a $w^{1118}$ background. Other GFP, RFP, and mCherry lines have been described previously (see Key Resources Table).

For the *Spd-2* mutant embryo analyses we used embryos laid by *spd-2$^{Z35711}$/spd-2 $^{Df(3L)st-j7}$* or *spd-2$^{Z35711}$/spd-2$^{G20143}$* transheterozygotes expressing two copies of the Spd-2-GFP fusions, or one copy of a Spd-2 fusion and one copy of another fusion protein. *Drosophila melanogaster Oregon-R* and $w^{67}$ were used as a WT stock where indicated. Balancer chromosomes and markers used were described previously (FlyBase, USA).

## Centrosome purification

Whole centrosomes were isolated from extracts of early *Drosophila* embryos (0–4 hr) using a modified version of a centrosome isolation protocol (*Lehmann et al., 2006*). Embryo extract containing 50% sucrose was layered on top of a sucrose cushion comprising 55% and 70% sucrose. The tubes were spun at 27,000 rpm, causing the centrosomes in the extract to move through the 55% layer and into the 70% sucrose layer. A 'Cytosolic' fraction was collected from the top of the tube, and fractions were then collected from the bottom of the tube. Western blotting was performed to identify the 'Centrosome' fractions that contained the greatest enrichment of centrosomal proteins. Phosphatase treatment was carried out on the centrosome fractions using alkaline phosphatase (Roche) for 4.5 hr at 37°C with or without phosphatase inhibitor cocktails 2 and 3 (Sigma).

## Centrosome immunoprecipitation and mass spectrometry

Centrosomes were immunoprecipitated from the centrosomal fractions using anti-Cnn antibodies coupled to protein A conjugated magnetic Dynabeads (Life Technologies). Cytoplasmic Spd-2 was immunoprecipitated from the cytoplasmic fractions using anti-Spd-2 antibodies coupled to Dynabeads. The dynabead/antibody suspensions were rotated at 4°C overnight. The antibodies were cross-linked to the beads using the BS3 crosslinker (Thermo Fisher). Centrosomal and cytoplasmic fractions were diluted 1:1, added to the antibody-crosslinked beads and rotated at 4°C for 2 hr. Beads were washed, boiled in sample buffer (SB) and separated on a polyacrylamide gel, and the band containing Spd-2 was cut out. Samples were prepared for mass spectrometry and enriched for phosphopeptides as described previously (*Conduit et al., 2014a*). Liquid chromatography-MS/MS analysis was performed using a LTQ Orbitrap Mass Spectrometer (Thermo Scientific) coupled to an UltiMate 3000 Nano LC system (Thermo Scientific). The mass spectrometry data were searched against the FlyBase sequence database (http://flybase.bio.indiana.edu/) using Mascot software (Matrix Science). The following settings were used for the searches: enzyme: trypsin; fixed modification: carbamidomethylation; variable modifications: methionine oxidation, glutamine/asparagine deamidation; serine/threonine/tyrosine phosphorylation; error tolerance for the precursor ions, 20

ppm; mass error tolerance for the fragment ions, 0.6 Da; number of missed cleavage sites, 3. The MS/MS spectra for identified phosphopeptides were manually inspected in Mascot.

## Recombinant protein expression and purification

The cDNA sequences encoding *Drosophila* Spd-2$_{352-758}$ (WT and ALL mutant) were subcloned into a pETM44 (EMBL) vector encoding an N-terminal His6-MBP tag. Proteins were expressed in *Escherichia coli (E. coli)* B21 strains in LB, and purified using a pre-poured amylose column containing 4 mL amylose resin (New England Biolabs) followed by size exclusion chromatography (protein buffer: 20 mM Tris pH 8.0, 150 mM NaCl, 0.5 mM TCEP) using an AKTA pure chromatography system with a HiLoad-Superdex 200 16/600 column attached (GE Healthcare).

## In vitro interaction assays

Anti-MBP antibody was coupled to magnetic beads (7.5 µg of antibody per 1 mg of beads) using the Dynabeads Antibody Coupling Kit (Thermo Fisher), following manufacturer's instructions. Each sample (100 µL of resuspended beads) was incubated with 32.2 µg of the appropriate protein in protein buffer (see above) for 30 min rotating at RT. The beads were rinsed twice with kinase buffer (CST) and resuspended in 60 µL of kinase buffer containing 200 µM of ATP (CST) and either kinase storage buffer (50 mM HEPES pH 7.6, 100 mM NaCl, 5 mM DTT, 20% glycerol, 15 mM reduced glutathione), for non-phosphorylated 'blank' controls; or 8.8 ng/µL of commercial PLK1 kinase (ProQinase) for phosphorylated samples. The samples were rinsed 3X with binding buffer (50 mM Tris pH 8.0, 200 mM NaCl, 1 mM DTT, 0.1% Tween-20, 10 mg/mL BSA, 1X phosphatase inhibitor cocktails 2 and 3, 1X SIGMAFAST EDTA-free protease inhibitor cocktail (Sigma)). The beads were then resuspended in 0.3 mL of 0.2 µM GST-Plk1-PBD (Sigma) in binding buffer, and incubated rotating for 3 hr at 4 °C. The beads were then rinsed 3X with 500 µL of bead wash buffer B (50 mM Tris pH 8.0, 200 mM NaCl, 1 mM DTT, 0.1% Tween-20, 1 mg/mL BSA, 1X phosphatase inhibitor cocktails 2 and 3, 1X SIGMAFAST EDTA-free protease inhibitor cocktail), transferred to a clean tube, and rinsed once with bead wash buffer HA (same as wash buffer B, but without BSA) before protein elution with 1X SB.

## Western blot analysis

Western blotting to estimate embryonic protein levels was performed as described previously (*Novak et al., 2014*). The following primary antibodies were used for western blot analysis (see Key Resources Table): rabbit anti-Spd-2 (1:500), rabbit anti-Cnn (1:1000), mouse anti-γ-tubulin (1:500), mouse anti-Actin (1:2000), mouse anti-GST (1:500) and rabbit anti-GAGA factor (1:500). HRP conjugated secondary antibodies used (all at 1:3000): swine anti-rabbit (Dako), or ECL anti-mouse and ECL anti-rabbit (GE Healthcare).

## RNA synthesis and microinjection

The mRNA injection assay and the modified pRNA destination vector with the C-terminal mKate2 tag used here have been described previously (*Novak et al., 2014*; *Novak et al., 2016*). Spd-2-ALL and Spd-2-CONS cDNA were introduced into the vector via Gateway cloning. Two point mutations were introduced into WT Spd-2-mKate2 using QuikChange mutagenesis (Agilent) to generate Spd-2-AA-mKate2. Spd-2-NT-mKate2 and Spd-2-CT-mKate2 partial mutants were derived from PCR-amplified fragments of WT Spd-2-mKate2 and Spd-2-ALL-mKate2 via NEBuilder HiFi assembly (New England Biolabs). The last potential binding site mutated in the N-terminus group was S538-S540, and the first potential binding site mutated in the C-terminus group was S581-S582, so that each group would include 17 potential sites. In vitro RNA synthesis was performed using a T3 mMESSAGE mMACHINE kit (Thermo Fisher) and RNA was purified using an RNeasy MinElute kit (Qiagen). All RNA constructs were injected at a concentration of 2 mg/mL.

## Immunofluorescence

Embryos were collected for 0–1 hr, aged for 45–60 min, and processed as described (*Stevens et al., 2010*). Samples were mounted onto microscopy slides with high-precision glass coverslips (CellPath). Specifics for each experiment as follows:

## Quantification of successful completion of pronuclear fusion
Embryos were stained using a mouse anti-α-tubulin (1:1000), followed by Alexa 594 nm anti-mouse and GFP-Booster Atto488 (1:500 dilution). Samples were mounted in Vectashield medium with DAPI. Embryos were counted using a Zeiss Axioskop two microscope (Zeiss International) with a 10x/0.30-NA and a 40x/0.75-NA objectives. Embryos were counted as developing beyond pronu-clear fusion if they had clearly reached syncytial/gastrulation stages. For each of the four conditions (non-rescue, WT-rescue, CONS-rescue and ALL-rescue), we performed two biological replicates (embryos from separate sets of mothers), each with three technical replicates (embryos collected and processed independently);>50 embryos were counted per sample.

## Phospho-Cnn staining
Embryos were stained using a guinea pig anti-Cnn antibody (1:1000) and a rabbit anti-Cnn pSer567 antibody (1:500); followed by Alexa 594 nm anti-rabbit, CF405S anti-guinea pig, and GFP-Booster Atto488 (1:500 dilution). Samples were mounted in Vectashield medium without DAPI.

## γ-tubulin staining
Embryos were stained using a mouse anti-γ-tubulin antibody (1:500), followed by Alexa 594 nm anti-mouse and GFP-Booster Atto488 (1:500 dilution). Samples were mounted in Vectashield medium with DAPI.

## Imaging
### Spinning disk confocal microscopy
Embryos were imaged at 21°C on a Perkin Elmer ERS spinning disk (Volocity software) mounted on a Zeiss Axiovert 200M microscope using a 63X/1.4-NA oil immersion objective and an Orca ER CCD camera (Hamamatsu Photonics, Japan). 488- and 561 nm lasers were used to excite GFP and RFP/mCherry, respectively. Confocal sections of 13 slices with 0.5-μm-thick intervals were collected every 30 s (17 slices for the analysis of protein dynamics throughout the cell cycle). Focus was occasionally manually readjusted in between intervals.

### 3D-SIM
3D-SIM microscopy was performed and analysed as described (*Conduit et al., 2014a*) on an OMX V3 Blaze microscope (GE Healthcare, UK) with a 60x/1.42-NA oil UPlanSApo objective (Olympus); 405-, 488- and 593 nm diode lasers, and Edge 5.5 sCMOS cameras (PCO). The raw acquisition was reconstructed using softWoRx 6.1 (GE Healthcare) with a Wiener filter setting of 0.006 and channel-specific optical transfer function. Living embryos were imaged at 21°C, acquiring stacks of 6 z-slices (0.125 μm intervals). Stacks of 13 z-slices (0.125 μm intervals) were acquired from fixed samples (phospho-Cnn staining). The images shown are maximum intensity projections. For multi-colour 3D-SIM, images from the different colour channels were registered with alignment parameters obtained from calibration measurements using 1 μm to 0.2 μm TetraSpeck Microspheres (Thermo Fisher) using OMX Editor and Chromagnon alignment software. The SIM-Check plug-in in ImageJ (NIH) was used to assess the quality of the SIM reconstructions (*Ball et al., 2015*).

### Airyscan
Fixed samples (γ-tubulin staining) were imaged using an inverted Zeiss 880 microscope fitted with an Airyscan detector. The system was equipped with Plan-Apochromat 63x/1.4-NA oil lens. The laser excitation lines used were 405 nm diode, 488 nm argon and 561 nm diode laser. Stacks of 25 slices with 0.14-μm-thick intervals were collected with pixel size (xy) of 0.035 μm, using a piezo-driven z-positioner stage. Images were Airy-processed in 3D with a strength value of 'auto' (~6). The software used to acquire images and process the images taken in super-resolution Airyscan mode was ZEN (black edition, Zeiss).

## Image and statistical analysis

### Blind analysis of 3D-SIM images

Centrosome images were selected based on quality of the reconstruction as assessed by the SIM-Check plug-in and the presence of a visible, well-formed ring corresponding to the presence of protein at the mother centriole wall. Each individual centrosome image was saved as a separate file, renamed and randomised post acquisition. The entire dataset for each experiment were scored blind by researchers not involved in any aspect of the data acquisition. The Spd-2-GFP in mutant background dataset was scored by one person. It included three different conditions (WT, CONS and ALL) with 36 centrosomes per condition. The Spd-2-GFP in WT background and Spd-2-mCherry/Polo-GFP datasets included three and two conditions (WT, CONS and ALL; or WT and CONS), respectively. The former included 32 images per condition, and the latter included 30 images per condition (16 and 15 individual centrosomes, two different channels, respectively). They were scored independently by three different people, and an average score was calculated. The Spd-2-mKate2 injection into PoloGFP dataset included two conditions (WT and CONS) with 20 and 22 images each (10 and 11 individual centrosomes, two different channels) and it was scored by one person.

### Analysis of centrosome and MT fluorescent intensities

We used ImageJ to calculate the maximum intensity projection of z-stacks of movies taken from the PE spinning disk system. The time frame chosen for analysis corresponded to 1 min before nuclear envelope breakdown. The five brightest centrosomes per embryo were identified via manual thresholding and analysed; the number of embryos analysed is indicated in each Figure. For both the green and red channels we measured the mean intensity within a square of fixed size (5.04 μm x 5.04 μm) centred manually on each individual centrosome. Similarly, we measured the mean intensity of the background near each centrosome. We calculated the average centrosome intensity and subtracted the average background intensity per embryo. The values for all the embryos were plotted on Prism 7 (GraphPad Software). Prism was also used to check column statistics and Gaussian distribution of the data. For the Jupiter-mCherry/Spd-2-GFP data we used the D'Agostino–Pearson omnibus normality test. For the statistical analysis, we used ordinary one-way ANOVA with Tukey's multiple comparisons test if data passed the normality test, or the Kruskal-Wallis test with Dunn's multiple comparisons test otherwise. For the dataset comparing Spd-2-GFP and Spd-2-11A-GFP we used the Shapiro-Wilk normality test followed by the unpaired t test with Welch's correction. Significance in statistical tests was defined by $p < 0.05$.

### Radial profiling of centrosomes

We used ImageJ to calculate an average 'radial profile' of the distribution of the different PCM proteins around the mother centriole (*Conduit et al., 2014b*). For embryos imaged live, the five brightest centrosomes in each embryo were analysed (the number of embryos analysed for each genotype is indicated in the individual Figures). For the analysis of fixed embryos, we analysed 1 pair of centrosomes per embryo, five embryos per technical replicate (embryos collected and processed independently), and three technical replicates in total per condition (so total centrosomes analysed = 15).

For each individual centrosome we found its center of mass by thresholding the image and running the 'analyse particles' (centre of mass) macro on the most central Z plane of the centrosome, as described (*Conduit et al., 2014b*). We then centred concentric rings spaced at 0.021 μm and spanning across 2.09 μm on this centre (0.007 μm and spanning across 1.41 μm for the fixed γ-tubulin images) and measured the average fluorescence in each ring, and subtracted the average cytosolic signal. Each individual centrosome profile was then normalised to the average peak intensity for all the centrosomes of the control condition (WT Spd-2-GFP embryos). Each profile was then mirrored to produce a full centrosome profile. The final radial profiles shown are an average of all the full centrosomal profiles per condition. In some graphs, we show a 'normalised' profile, where each individual centrosome profile was normalised to the average peak intensity of its corresponding condition (rather than to the WT Spd-2-GFP embryo control). The resulting radial profile peaks for all conditions were then normalised to 1; this allows the distribution of different proteins around the centriole to be compared, independently of differences in centrosomal protein levels.

## Analysis and regression modelling for the dynamics of Spd-2-GFP

Spd-2-GFP dynamics throughout the cell cycle were analysed as described (*Aydogan et al., 2018*). Briefly, we used ImageJ to calculate the maximum intensity projection of z-stacks of movies taken from the PE spinning disk system. The backgrounds were subtracted using the subtract background function with a rolling ball radius of 10 pixels. Spd-2-GFP foci (centrosomes) were tracked using TrackMate plug-in (*Tinevez et al., 2017*) with the following analysis settings: track spot diameter size of 2.1 μm, initial threshold of 0, and quality of >0.03. Regression analysis on the centrosome growth curves were carried out using Prism, and the mathematical modelling was done using the nonlinear regression (curve fit) analysis function, excluding the last three points (90 s) of the cell cycle (as the data was very variable towards the end of mitosis).

The data for WT Spd-2-GFP (*Figure 8*, *blue line*) was fitted against four different functions to assess the most suitable model (*Figure 8—figure supplement 3A*): (1) linear growth followed by linear decrease; (2) linear growth followed by plateau followed by linear decrease; (3) Gaussian function; (4) Lorentzian function. Functions (1) and (2) are bespoke algorithms, with the following equations for $X$ amount of time:

$$
\begin{aligned}
Y_1 &= intercept_1 + slope_1 * X \\
Y_{X0} &= slope_1 * X_0 + intercept_1 \\
Y_2 &= Y_{x0} + slope_2 * (X - X_0) \\
Y &= IF(X{<}X_0, Y_1, Y_2)
\end{aligned}
\tag{1}
$$

$$
\begin{aligned}
Y_1 &= intercept_1 + slope_1 * X \\
Y_{X0} &= slope_1 * X_0 + intercept_1 \\
Y_2 &= Y_{X0} + slope_2 * (X - X_0) \\
slope_2 &= 0 \\
Y_{X1} &= Y_{X0} + slope_2 * (X_1 - X_0) \\
Y_3 &= Y_{X1} + slope_3 * (X - X_1) \\
Y &= IF(X{<}X_0, Y_1, IF\ (X{<}X_1, Y_2,\ Y_3))
\end{aligned}
\tag{2}
$$

The only constraints applied to these equations were the requirements for $slope_1$ and inflection points ($X_0$, $X_1$) to be greater than 0, and $slope_3$ to be less than 0. Centrosomes that come from a single embryo were treated as internal replicates, and thus the fitting used only the mean $Y$ value of each time point. To judge and control the quality and precision of regression (goodness-of-fit), we used the $R^2$, adjusted $R^2$, and absolute sum-of- square values. To compare the fits, the extra sum-of-squares F test was applied, and the appropriate fit was chosen by selecting the simpler model unless $p{<}0.05$. The 'linear growth followed by plateau followed by linear decrease' model best fit the data (*Figure 8—figure supplement 3A*), but it is likely a simplification of a more complex model, so individual curves are shown for each embryo without any model fitted (average of >44 centrosomes per embryo) (*Figure 8—figure supplement 1*).

The data for Spd-2-CONS-GFP (*Figure 8*, *red line*) was fitted against the 'linear growth followed by plateau followed by linear decrease' model, as this was the preferred model for the WT data. As this data seemed to better be described as a straight line, this model was also compared to a standard straight line model (with no slope constraints)—function (5); or a user-defined constant line function (6) (*Figure 8—figure supplement 3B*):

$$
Y = intercept_1 + 0 * X
$$

The preferred model was the straight line—function (5) (*Figure 8—figure supplement 3B*)—although the average slope value was nearly zero ($-0.0005 \pm 0.0054$; mean $\pm$ SD), indicating that the appropriate model in practice would be a constant line.

## FCS (fluorescence correlation spectroscopy)

FCS measurements were obtained as previously described (*Aydogan et al., 2019*). For all measurements, the laser power was kept constant at 6.31 μW, and the heating unit of the microscope at 25 ° C. All autocorrelation functions (ACFs) were fitted with the eight previously described models, and

**Table 3.** Fitting parameters and chosen diffusion models for FCS experiments.

$\alpha$ , Anomalous subdiffusion parameter; AR, Structural parameter; ds, Diffusing species; bs, Blinking state of the fluorophore; ts, Triplet state of the fluorophore.

| Protein | Fitting boundaries (ms) | $\alpha$ | AR | Model |
|---------|------------------------|-----------|-----|-------|
| mGFP | $4 \times 10^{-4}$ - $1 \times 10^{2}$ | 1.00 | 5 | one ds one ts |
| Spd-2-GFP | $4 \times 10^{-4}$ - $1 \times 10^{3}$ | 0.65 | | one ds one ts |
| mNeonGreen (pPlk4) | $4 \times 10^{-4}$ - $2 \times 10^{2}$ | 0.70 | | one ds one ts |
| mNeonGreen (pSas-6) | $4 \times 10^{-4}$ - $2 \times 10^{2}$ | 0.75 | | one ds one ts |
| dNeonGreen | $4 \times 10^{-4}$ - $4 \times 10^{2}$ | 0.85 | | one ds one bs one ts |

DOI: https://doi.org/10.7554/eLife.50130.051

the best model was chosen based on the Bayesian information criterion. All fitting parameters and chosen diffusion models are stated in *Table 3*.

Purified monomeric Spycatcher-GFP (kind gift from A. van der Merwe and colleagues) was measured in vitro in 1x PBS + 0.05% Tween20 at a similar concentration as Spd2-GFP in vivo. Embryos from mothers expressing GFP-tagged Spd2 were measured at the centrosomal plane and in nuclear cycles 11–14 at the beginning of S-phase.

The molecular brightness of the measured fluorophore (presented as photon count-rate per molecule (CPM)) was used to identify the oligomerization state of cytoplasmic, fluorescently-tagged Spd2 (in comparison to monomeric GFP). The CPM measurements were corrected for background fluorescence (with 10 control measurements of empty buffer for in vitro experiments, and ~20 recordings from WT embryos for in vivo experiments) using the following equation:

$$CPM = \frac{\left(Photon\ count\ rate^{SAMPLE} -\ Photon\ count\ rate^{CONTROL}\right)}{Number\ of\ particles\ in\ observation\ spot}$$

In addition, control lines from mothers expressing monomeric or dimeric NeonGreen were measured to test the sensitivity of our CPM-based analysis. Identical to Spd2-GFP measurements, the recordings were taken within nuclear cycles 11–14 and at the beginning of S-phase. These flies also expressed Asl-mKate2 expressed from its endogenous promoter to identify the correct nuclear cycle stage and the centrosomal plane.

# Acknowledgements

We thank the members of the Raff Laboratory for critically reading the manuscript and stimulating discussions. We are grateful to Anna Caballe and Lisa Gartenmann for experimental advice; Omer Dushek for advice regarding statistical analysis; Francis Barr and Ricardo Nunes Bastos for help with the phosphopeptide enrichment and the in vitro kinase assays; Ben Thomas for mass spectrometry assistance and advice; Jonathan Bohlen for help with the radial profile analysis; Anton van der Merwe and his lab for the kind gift of purified monomeric Spycatcher-GFP for FCS analysis; Michael Barton and Andreas Haensele for taking part in the blind scoring and Andreas Haensele for help with protein purification; and the members of Micron Oxford for help and advice on live 3D structured illumination microscopy.

Superresolution microscopy was performed at the Micron Oxford Advanced Bioimaging Unit, funded by a Strategic Award from the Wellcome Trust (107457). The research was funded by a Wellcome Trust Senior Investigator Award (104575; T L Steinacker, S Saurya, A Wainman, ZA Novak, PT Conduit and JW Raff), a Wellcome Trust PhD studentship (I Alvarez-Rodrigo) and an Edward Penley Abraham Scholarship (to MG Aydogan and J Baumbach).

## Additional information

### Funding

| Funder | Grant reference number | Author |
|---|---|---|
| Wellcome | Wellcome Trust Senior Investigator Award, 104575 | Thomas L Steinacker Saroj Saurya Paul T Conduit Zsofia A Novak Alan Wainman Jordan W Raff |
| Wellcome | Wellcome Trust PhD studentship, 109096 | Ines Alvarez-Rodrigo |
| Wellcome | Wellcome Trust Strategic Award, 107457 (partially supported) | Alan Wainman |
| Edward Penley Abraham Scholarship | Graduate Student Scholarship | Janina Baumbach Mustafa G Aydogan |

The funders had no role in study design, data collection and interpretation, or the decision to submit the work for publication.

### Author contributions

Ines Alvarez-Rodrigo, Conceptualization, Resources, Formal analysis, Validation, Investigation, Visualization, Writing—original draft, Writing—review and editing; Thomas L Steinacker, Conceptualization, Resources, Formal analysis, Validation, Investigation, Visualization, Methodology; Saroj Saurya, Resources, Investigation; Paul T Conduit, Formal analysis, Validation, Investigation, Visualization, Methodology; Janina Baumbach, Resources, Formal analysis, Validation, Investigation, Visualization; Zsofia A Novak, Conceptualization, Resources, Formal analysis, Supervision, Methodology, Writing—review and editing; Mustafa G Aydogan, Formal analysis, Methodology, Writing—review and editing; Alan Wainman, Formal analysis, Supervision, Writing—review and editing; Jordan W Raff, Conceptualization, Supervision, Funding acquisition, Visualization, Methodology, Writing—original draft, Project administration, Writing—review and editing

### Author ORCIDs

Ines Alvarez-Rodrigo (iD) https://orcid.org/0000-0003-2181-5535
Thomas L Steinacker (iD) https://orcid.org/0000-0002-7244-5610
Saroj Saurya (iD) https://orcid.org/0000-0003-4057-0123
Paul T Conduit (iD) https://orcid.org/0000-0002-7822-1191
Mustafa G Aydogan (iD) https://orcid.org/0000-0003-1673-0596
Alan Wainman (iD) http://orcid.org/0000-0002-6292-4183
Jordan W Raff (iD) https://orcid.org/0000-0002-4689-1297

### Decision letter and Author response

Decision letter https://doi.org/10.7554/eLife.50130.057
Author response https://doi.org/10.7554/eLife.50130.058

## Additional files

### Supplementary files

• Supplementary file 1. Key Resources Table.
DOI: https://doi.org/10.7554/eLife.50130.052
• Transparent reporting form
DOI: https://doi.org/10.7554/eLife.50130.053

## Data availability

No major datasets have been generated for this study. All data generated or analysed during this study are included in the manuscript and supporting files. As part of this study, we consulted the publicly available iProteinDB database; the corresponding papers have been cited in the main text of this article and further details are provided below. Source data files have been provided for Figure 1 and Figure 1-figure supplement 1D; Figure 2 and Figure 2-figure supplement 1; Figure3B,C; Figure 4; Figure 5; Figure 5-figure supplement 1; Figure 6; Figure 7C,D; Figure 7-figure supplement 1A,B and Figure 7-figure supplement 1E,F; Figure 8, Figure 8-figure supplement 1 and Figure 8-figure supplement 2; Figure 9B-E; Figure 9-figure supplement 1 and Figure 10.

The following previously published dataset was used:

| Author(s) | Year | Dataset title | Dataset URL | Database and Identifier |
|---|---|---|---|---|
| Hu Y, Sopko R, Chung V, Foos M, Studer RA, Landry SD, Liu D, Rabinow L, Gnad F, Beltrao P, Perrimon N | 2019 | iProteinDB: An Integrative Database of Drosophila Post-translational Modifications (FBpp0075122) | https://www.flyrnai.org/tools/iproteindb/web/protein/FBpp0075122/ | iProteinDB, FBpp0075122 |

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
