## [Decision Letter]

[Editors’ note: a previous version of this study was rejected after peer review, but the authors submitted for reconsideration. The first decision letter after peer review is shown below.]

Thank you for submitting your work entitled "A positive feedback loop drives centrosome maturation in flies" for consideration by *eLife*. Your article has been reviewed by three peer reviewers, and the evaluation has been overseen by a Reviewing Editor and a Senior Editor. The reviewers have opted to remain anonymous.

Our decision has been reached after consultation between the reviewers. Based on these discussions and the individual reviews below, we regret to inform you that your work will not be considered further for publication in *eLife*.

As you can see in the individual reviewers' comments, all appreciated how well the study is done addressing a critical question in the field of centrosome biology. However, all remained concerned that the study did not provide a proof of Spd-2-Polo interaction being critical. They felt that proving this point is not necessarily straightforward in the scope of a simple revision (leaving the possibility open that those experiments disprove the model, or it might be not straightforward to design the experiments to prove this point).

With that said, all reviewers are enthusiastic about this study overall: although we cannot consider publication of your manuscript at this point per *eLife*'s policy to invite revision only if the revision experiments are straightforward and the results are unlikely to change the major conclusions, we would be happy to reconsider your work as a new submission, if you're able to address the major points of reviewers.

Reviewer #1:

The manuscript "A positive feedback loop drives centrosome maturation in flies" by Alvarez-Rodrigo et al. investigates the regulation of centrosome maturation when cells prepare for mitosis, by focusing on the expansion of the pericentriolar material (PCM). In particular, the authors analyze the involvement of Spd-2 and Polo in this process and study potential phosphorylation sites in Spd-2 that may help recruit Polo and that seem to be important for pericentriolar scaffold assembly and expansion, including the recruitment of other essential PCM proteins such as γ-tubulin to the expanded scaffold. They propose that Spd-2 is an essential component of a feedback mechanism that involves recruitment of Polo to drive incorporation and phosphorylation of Cnn near centrioles and subsequent outward movement of Cnn-Spd-2, thereby contributing to assembly of an expanded PCM. This process is driven by continuous Spd-2-Polo-dependent incorporation of additional Cnn near centrioles and its outward flux.

Overall this is a very nice story that puts together various previous and new observations into an appealing model to explain the mechanism by which centrosomes mature, at least in flies.

The paper is very well written and the data are presented clearly.

However, I found one major issue with this work that needs to be addressed. In my opinion the authors do not provide convincing evidence for a core element of their model: the role of Spd-2 in Polo recruitment.

Specific points:

1) The authors state "Based on our observations below, we refer to these mutant proteins as being unable to recruit Polo." However, the recruitment of Polo to Spd-2 is never demonstrated in the first place. I understand that the interaction is supposed to happen specifically at the centrosome, so immunoprecipitation from soluble extract is most likely not an option, but can the author provide any evidence for a direct interaction of Polo with Spd-2?

2) The observations that follow the authors' statement do not demonstrate the inability of the mutants to recruit Polo: it is shown that the mutants partially rescue embryos lacking Spd-2, that mutants are less concentrated at centrosomes, that these centrosomes have less MTs, and that mutants and Polo localize around centrioles but do not assemble into an expanded pericentriolar scaffold. This latter finding is the only piece of data in support of the proposed role of Spd-2 in Polo binding. However, as the authors also state, in these experiments the scaffold may simply not assemble. In this case the reduced Polo signal could simply be the result of the absence of an expanded scaffold and not necessarily due to lack of binding to Spd-2.

3) The demonstration that Spd-2 is phosphorylated at centrosomes is very convincing (Figure 1). However, it was not tested whether this phosphorylation is at least partially Polo-dependent. This could indicate interaction with Polo at centrosomes.

4) Figure 2: Are the mutants still phosphorylated at centrosomes? This should be revealed by an analysis as in Figure 1.

*Reviewer #2:*

In the manuscript by Alvarez-Rodrigo et al. entitled "A positive feedback loop drives centrosome maturation in flies" the authors aim to show that the interactions between Spd2-Polo-Cnn drive an autocatalytic expansion of the PCM to ensure synchronized growth of sister centrosomes. Based on a minimal Polo box binding consensus sequence, the authors mutated Spd2 to prevent Polo binding and then characterized the effects on fly viability, Spd2 localization, and PCM expansion. They found that increasing the mutational burden on Spd2 reduces PCM expansion and localization of centrosome components during centrosome maturation. If the authors were able to show the mechanism in Figure 10, this would constitute a major contribution to the centrosome field. Unfortunately, the lack of justification or biological relevance behind the authors' approach prevent them from achieving this goal. Therefore, I do not believe the manuscript should be published without extensive changes to the approach or substantial work to justify and validate their approach.

The authors perform initial MS/MS to identify the phosphorylated residues on Spd2 which may be a prerequisite for Polo binding. They then mutated these residues and found mild Spd2 defects that did not recapitulate the phenotypes found in the CONS or ALL mutants. From these data, it appears that mutating the biologically relevant sites was not sufficient to produce a phenotype, so the authors moved on to mutating a substantial number of serine residues in Spd2 only because it appears in a minimal consensus sequence. This approach led to studying the effects of a massively mutated protein without regard to the biology of the system.

Furthermore, the authors do not provide evidence that the Spd2 mutants result in an inability of Polo to bind Spd2. Perhaps Polo can bind Spd2 through non-canonical sites? That may be the case because the authors present evidence that Polo is still recruited to the centriole and functions properly, as Cnn is phosphorylated and centrioles are still able to disengage properly. Are the Spd2 mutants still being phosphorylated, as in Figure 1? This can be answered with their established method in Figure 1 and by mass spectrometry. Is mutant Spd2 still able to homodimerize and interact with Cnn? The lab has previously established a yeast 2-hybrid assay and has used SEC-MALS, both of which could be used to test this. Spd2 also interacts with Asterless and PLP. Do the Spd CONS or ALL mutants disrupt these interactions, which may explain the mutant phenotypes? Further biochemical analysis of mutant Spd2 should be performed to show that the mutants do not cause protein misfolding.

Although their discussion and conclusions are intriguing and could provide insight into how centrosome maturation may be regulated, I do not feel they have provided sufficient evidence to support these conclusions. Additionally, while there is a striking phenotype in the manuscript, the authors have only shown that this is caused by mutating Spd2. They have not shown that this is due to disrupting the interaction with Polo, let alone disrupting a potential positive feedback loop, as implied by the title of their manuscript, which is misleading.

Additional concerns:

1) All experiments are performed in *Drosophila* embryos, whereas the authors claim that the mechanism is universal to the entire fly "in vivo". This should be toned down.

2) The manuscript would benefit from more details of the mitotic defects in Figure 2A.

3) Are Spd2ALL, Spd2^-/-^ heterozygotes viable or can reduced dosage explain the phenotypes in Figure 5—figure supplement 1?

4) The images of centrosomes in Figures 4-6 appear to be S-phase centrosomes with flares, not mitotic centrosomes. The authors reference mitosis and maturation when, in fact, they show images of interphase (S-phase) centrosomes. The authors should show low magnification images of mitotic spindles and then high magnification insets of centrosomes to demonstrate that they are, in fact, examining mitotic centrosomes.

*Reviewer #3:*

In this manuscript, the authors show that Spd-2 is partially phosphorylated at centrosomes, allowing the recruitment of Polo to the centrosomes. Disabling the recruitment leads to outward pericentriolar scaffold expansion failure, and prevents centrosome maturation. The authors also demonstrate that Polo, Spd-2 and Cnn together can drives mitotic centrosome expansion by forming a positive feed-back loop. This is an interesting paper with data of high quality. I note the following issue the authors need to consider:

1) The authors should provide direct evidence about how much Spd-2-CONS and Spd-2-ALL are affects their binding with the Polo PBD compared to WT, which is critical for most of their conclusions. I'm concerned that mutating all 34 Serine sites may affect protein structure and this needs to be ruled out.

2) Do the authors now if the phosphorylation status is altered during the cell cycle?

3) Figure 4: authors conclude that the inability of mutant proteins to recruit Polo reduces the assembly and/or maintenance of this mixed scaffold, but the data shown can't exclude the possibility that mutant proteins are efficiently incorporated into the scaffold. The author should show a comparison of endogenous Spd-2 scaffold in all three conditions.

4) Figure 5: I'm not convinced by the authors' conclusion that the pericentriolar scaffold cannot expand because Spd-2-CON can't recruit Polo. First, they co-localize at the mother centriole, and second, Spd-2-CON itself is affecting scaffold formation in Spd-2 mutant background (Figure 4), so how can they be certain it's a Polo recruitment issue? I would potentially suggest a FRAP experiment to showcase the recruitment of Polo to the scaffold. In addition, apart from co-expressing with Polo-GFP, what's the difference between Figure 5B and Figure 4A (middle lane)? Why is there's a huge difference between the ratios (64/36 vs 9/91)? This was confusing to me.

5) Figure 7: what is the component of the scaffold? If the scaffold is made of Spd-2, when Spd-2 CONS is unable to form the scaffold, you won't be able to see other proteins there no matter how they are recruited. Is there any other marker for the scaffold?

6) Figure 9: The C-terminus of Spd-2 contains most of the conserved Serine sites, but the authors show that mutation N-terminal sites have the biggest effect? What do the authors think the molecular basis for this is?

[Editors’ note: what now follows is the decision letter after the authors submitted for further consideration.]

Thank you for submitting your article "Evidence that a positive feedback loop drives centrosome maturation in fly embryos" for consideration by *eLife*. Your article has been reviewed by three peer reviewers, and the evaluation has been overseen by a Reviewing Editor and Anna Akhmanova as the Senior Editor. The reviewers have opted to remain anonymous.

The reviewers have discussed the reviews with one another and the Reviewing Editor has drafted this decision to help you prepare a revised submission.

Summary:

In the manuscript by Alvarez-Rodrigo et al., the authors show that the interactions between Spd2-Polo-Cnn drive an autocatalytic expansion of the PCM to ensure synchronized growth of sister centrosomes.

This is a resubmission of previously rejected manuscript, where all the reviewers appreciated the rigor of the work and an elegant model to explain the process of centrosome maturation but found that the direct evidence to support the importance of Spd-2-Polo interaction was lacking.

This revised and newly submitted version was reviewed by original reviewers, and now they all agree that this work now has strong evidence that supports the conclusions. Some reviewers raised a few minor comments, which the authors may want to address prior to formal acceptance.

Overall this is a very nice story that puts together various previous and new observations into an appealing model to explain the mechanism by which centrosomes mature, which may be conserved in other species as well.

*Reviewer #1:*

In the manuscript by Alvarez-Rodrigo et al. entitled "A positive feedback loop drives centrosome maturation in flies" the authors show that the interactions between Spd2-Polo-Cnn drive an autocatalytic expansion of the PCM to ensure synchronized growth of sister centrosomes. I feel that the authors have addressed my original concerns with one remaining issue. Regarding the immunoprecipitation (IP) experiments between the MBP-phospho-Spd2 fragment and GST-PBD (new Figure 1C), the resolution of these immunoblots appear low and the brightness/contrast altered such that it is difficult to gauge the load between the samples. It appears that there is more GST-PBD in the MBP-Spd-2 WT immunoprecipitate, but how much more? Replacing these with higher resolution blots with accompanying quantitation would be appropriate. This is an important experiment/result because it demonstrates that the authors have in fact generated a Spd-2 Polo-binding mutant. Also, is the kinase being used human Plk1, whereas the Spd-2 and PBD proteins are fly? Whatever the species, the fact that the authors used proteins from different organisms should be stated in the Results section.

I also recommend a couple of text changes in this section. The authors state that binding of the 19T mutant was reduced to background levels. How do they know this is background? This could be real binding and, therefore, would remain at WT levels, not a reduced level. The authors end this section stating, "Thus, a fragment of Spd-2 can bind directly to the PBD when phosphorylated, and this binding is prevented when the S-S/T motifs are mutated to T-S/T". Based on the data, it looks to me that binding is not prevented. Instead, they should say that phosphorylation of Spd-2 increases PDB binding which, in my opinion, is what the data shows. The data also suggest that Polo activity primes it's binding to Spd-2 (as they state in the Discussion) but this important finding should also be emphasized here at the end of this Results section.

*Reviewer #2:*

In their revised manuscript the authors have addressed my concerns by providing new experiments and additional discussion/explanation.

My main concerns were the lack of evidence for interaction between Spd-2 and Polo and the conundrum that the reduced Polo signal at centrosomes containing Spd-2 mutants may be due to specific loss of interaction with Polo or simply due to the lack of any scaffold where Polo could be recruited.

The first concern was dealt with by providing in vitro biochemical interaction data (Figure 1C) and the second by performing a new type of quantification of the centrosomal Polo signal in cells expressing a mix of WT and mutant Spd-2, at a time point when Polo signal is just starting to become reduced (Figure 10). This approach allowed the authors to demonstrate reduced Polo signal even in cells where a Spd-2 PCM scaffold is still present. Although the data supports the authors' conclusion that Spd-2 mutants cause a specific loss of PCM associated Polo, the quantification would be more convincing if actual intensities would have been quantified for the Spd-2 scaffold and Polo signals rather than subjective classification into "weak" and "strong". This is because in contrast to a similar quantification in Figure 6, where an "all or nothing" effect is observed, here the relative intensities of PCM scaffold-associated Spd-2 vs Polo are crucial for the result. I would suggest to also adding quantification of intensities to the figure.

My other concerns were also addressed, not in all cases with experiments, but I recognize the technical challenges that have led to this decision.

I would now support publication.

*Reviewer #3:*

The revised version of the paper is much improved and the authors have addressed most of the issues raised in my initial review to the best of their abilities. The manuscript in my opinion is now suitable for publication in *eLife*.

---

## [Author Response]

[Editors’ note: the author responses to the first round of peer review follow.]

Reviewer #1:[…] Overall this is a very nice story that puts together various previous and new observations into an appealing model to explain the mechanism by which centrosomes mature, at least in flies.The paper is very well written and the data are presented clearly.However, I found one major issue with this work that needs to be addressed. In my opinion the authors do not provide convincing evidence for a core element of their model: the role of Spd-2 in Polo recruitment.Specific points:1) The authors state "Based on our observations below, we refer to these mutant proteins as being unable to recruit Polo." However, the recruitment of Polo to Spd-2 is never demonstrated in the first place. I understand that the interaction is supposed to happen specifically at the centrosome, so immunoprecipitation from soluble extract is most likely not an option, but can the author provide any evidence for a direct interaction of Polo with Spd-2?

To address this question, we have now purified an MBP-fusion containing a fragment of Spd-2 (aa352-758; the only recombinant fragment of Spd-2 that is relatively stable in our hands). This fragment contains 19 S-S/T residues and we show that it can bind to recombinant GST-PBD, but only when it has first been phosphorylated by recombinant Plk1 (New Figure 1C). Mutating the 19 S-S/T residues in this Spd-2 fragment to T-S/T abolishes PBD-binding. These findings provide strong support for our hypothesis that *Drosophila* Spd-2 can directly bind to the Polo-PBD when phosphorylated, and that this binding is prevented when the S-S/T sites are mutated to T-S/T.

2) The observations that follow the authors' statement do not demonstrate the inability of the mutants to recruit Polo: it is shown that the mutants partially rescue embryos lacking Spd-2, that mutants are less concentrated at centrosomes, that these centrosomes have less MTs, and that mutants and Polo localize around centrioles but do not assemble into an expanded pericentriolar scaffold. This latter finding is the only piece of data in support of the proposed role of Spd-2 in Polo binding. However, as the authors also state, in these experiments the scaffold may simply not assemble. In this case the reduced Polo signal could simply be the result of the absence of an expanded scaffold and not necessarily due to lack of binding to Spd-2.

To address this problem we have now injected mRNA encoding either WT or mutant forms of *Spd-2-mKate2* into embryos expressing Polo-GFP. The mRNA is gradually translated, and so the Spd-2-mKate2 fusions gradually overwhelm the endogenous untagged Spd-2. We then looked for embryos injected with the mutant mRNA where the loss of Polo from the centrosome was first becoming apparent, and compared the centrosomal recruitment of the mutant Spd-2-mKate2 and Polo-GFP to similarly staged controls (injected with WT Spd-2-mKate2 mRNA). This analysis, scored blind, revealed that mutant Spd-2-mKate2 could still be detected in an expanded PCM even when the amount of Polo recruited to the scaffold was severely reduced (New Figure 10). This is presumably because there is enough WT Spd-2 present in these embryos to form a scaffold, but the mutant Spd-2 protein present in the scaffold cannot recruit Polo. This strongly supports our conclusion that the mutant Spd-2 proteins cannot recruit Polo to the expanded PCM. Nevertheless, we agree that we cannot formally exclude the possibility that the mutant Spd-2 proteins can still bind Polo in some way, and we have toned down our statements on this point throughout the manuscript.

3) The demonstration that Spd-2 is phosphorylated at centrosomes is very convincing (Figure 1). However, it was not tested whether this phosphorylation is at least partially Polo-dependent. This could indicate interaction with Polo at centrosomes.

This is a good question, but one that is difficult to address: a lack of Spd-2 phosphorylation after Polo inhibition might be a direct consequence of Polo failing to phosphorylate Spd-2, or an indirect consequence of there being no expanded PCM (as Spd-2 is mostly phosphorylated in the PCM). We do now provide evidence that Polo can phosphorylate a fragment of Spd-2 in vitro to create PBD binding sites (New Figure 1C), consistent with the possibility that at least some of the phosphorylation of Spd-2 at centrosomes is Polo-dependent.

4) Figure 2: Are the mutants still phosphorylated at centrosomes? This should be revealed by an analysis as in Figure 1.

This is a good question, but we have not attempted to answer it as it would be a lot of work to collect enough mutant material to do the biochemistry, and we think the answer may not be very informative: the S-S/T to T-S/T substitutions may or may not interfere with the phosphorylation of these motifs, but will perturb PBD-binding regardless.

Reviewer #2:[…] If the authors were able to show the mechanism in Figure 10, this would constitute a major contribution to the centrosome field. Unfortunately, the lack of justification or biological relevance behind the authors' approach prevent them from achieving this goal. Therefore, I do not believe the manuscript should be published without extensive changes to the approach or substantial work to justify and validate their approach.The authors perform initial MS/MS to identify the phosphorylated residues on Spd2 which may be a prerequisite for Polo binding. They then mutated these residues and found mild Spd2 defects that did not recapitulate the phenotypes found in the CONS or ALL mutants. From these data, it appears that mutating the biologically relevant sites was not sufficient to produce a phenotype, so the authors moved on to mutating a substantial number of serine residues in Spd2 only because it appears in a minimal consensus sequence. This approach led to studying the effects of a massively mutated protein without regard to the biology of the system.

This statement is based on the premise that we initially identified *all* the biologically relevant phosphorylation sites in our MS screen but that, when mutating these sites gave only a mild phenotype, we simply mutated many more sites (based only on a minimal S-S/T consensus) to generate a stronger phenotype. We disagree that these experiments were performed “without regard to the biology”. First, it is widely accepted that MS approaches may not identify *all* of the relevant phosphorylation sites in a protein, and this is demonstrably the case for Spd-2, where 4 independent MS screens (3 from embryos) have identified 41 phosphorylation sites in fly Spd-2, some of which were identified in multiple screens, but many of which were not (as summarised only for the sites that are potential PBD binding motifs in Table 2). Second, we considered a large body of evidence when devising this strategy: (1) The S-S/T(P) motif is widely accepted as a PBD-binding motif and the PBD is widely considered to be essential for targeting Polo/Plk1 to centrosomes^1-7^; (2) All of the previously identified sites in worm, human and frog SPD-2/Cep192 that recruit Plk1 to centrosomes conform to this consensus^8-11^; (3) The S-S/T to T-S/T substitution is widely accepted to prevent PBD binding^5,6^; (4) Ser-to-Thr substitutions are considered to be very conservative in nature, and so unlikely to lead to protein misfolding.

Furthermore, the authors do not provide evidence that the Spd2 mutants result in an inability of Polo to bind Spd2. Perhaps Polo can bind Spd2 through non-canonical sites? That may be the case because the authors present evidence that Polo is still recruited to the centriole and functions properly, as Cnn is phosphorylated and centrioles are still able to disengage properly. Are the Spd2 mutants still being phosphorylated, as in Figure 1? This can be answered with their established method in Figure 1 and by mass spectrometry. Is mutant Spd2 still able to homodimerize and interact with Cnn? The lab has previously established a yeast 2-hybrid assay and has used SEC-MALS, both of which could be used to test this. Spd2 also interacts with Asterless and PLP. Do the Spd CONS or ALL mutants disrupt these interactions, which may explain the mutant phenotypes? Further biochemical analysis of mutant Spd2 should be performed to show that the mutants do not cause protein misfolding.

In the third paragraph the reviewer raises several points. First, they ask for evidence that the Spd-2 mutants cannot bind Polo. As discussed in point 1 of our response to reviewer #1, we now provide compelling evidence that Spd-2 interacts directly with the PBD when phosphorylated, and this interaction is abolished when the putative PBD-binding motifs are mutated (New Figure 1C). Second, the reviewer highlights that the Spd-2 mutant proteins still colocalise with Polo at centrioles, suggesting that the mutant proteins are still recruiting Polo to centrioles. We now clarify that we believe Spd-2 is only required to recruit Polo to the PCM, not to the centrioles. There is strong evidence that Polo can be recruited to centrioles independently of Spd-2 by phosphorylated S-S/T(P) motifs in centriole proteins such as Sas-4^12^. Third, the reviewer asks whether the Spd-2 mutant proteins are still phosphorylated. Please see point 4 in our response to reviewer #1.

We have taken two approaches to address this point. First, we tested whether Spd-2 is a homodimer in fly embryos. This is crucial as, if so, a largely misfolded mutant protein might still localise to centrioles and centrosomes if it can still homodimerize with the WT protein. Using Fluorescence Correlation Spectroscopy (FCS), we show that cytoplasmic Spd-2 is monomeric (New Figure 5—figure supplement 1)—as was also recently shown for SPD-2 in worm embryos^13^. Second, with this information in hand, we examined how WT and mutant Spd-2-GFP fusion-proteins are recruited to centrioles in the presence of WT Spd-2-mCherry (which serves as a marker for the WT scaffold and also allows a scaffold to assemble even in the presence of the mutant proteins). The recruitment of the WT and mutant Spd-2-GFP fusions to the WT Spd-2-mCherry scaffold (scored blind) was essentially indistinguishable (New Figure 5). Thus, the mutant Spd-2 proteins can interact with all the proteins that recruit and maintain Spd-2 at centrioles and centrosomes, and this is unlikely to be because they are simply homodimerizing with the WT protein. Finally, we more clearly explain the significance of the data showing that the mutant Spd-2 proteins can rescue the defect in pronuclear fusion in embryos lacking Spd-2 (Figure 2C), but that these embryos still die of mitotic defects during early embryo development. This suggests that the mutant proteins allow centrioles to assemble sufficient PCM to promote the relatively slow process of pronuclear fusion, but not enough PCM to support the very rapid rounds of mitosis that follow. Taken together, these data suggest that the mutant proteins are not misfolded.

Additional concerns:

*1) All experiments are performed in Drosophila embryos, whereas the authors claim that the mechanism is universal to the entire fly "*in vivo*". This should be toned down.*

The reviewer points out that all our results are from fly embryos and may not apply to other cell types. We now make this clear throughout the manuscript.

2) The manuscript would benefit from more details of the mitotic defects in Figure 2A.

As requested, we now comment in more detail about the nature of the mitotic defects shown in Figure 2A (now Figure 3A).

3) Are Spd2ALL, Spd2^-/-^ heterozygotes viable or can reduced dosage explain the phenotypes in Figure 5—figure supplement 1?

The reviewer asks whether Spd2-ALL; Spd-2 ^-/-^ heterozygotes are viable, or whether reduced dosage can explain the phenotypes shown in Figure 5—figure supplement 1 (now Figure 6—figure supplement 1). These heterozygous flies are indeed viable, but this is to be expected as *Spd-2*^-/-^ mutants are also viable (but female sterile). We suspect the reviewer is suggesting that because there is only one copy of *Spd-2-ALL* in these flies, the phenotypes we observe may be due to lowering the genetic dosage of *Spd-2* rather than to the mutations we introduce into the genes. We think this is unlikely for two reasons: (1) In all the experiments where we assess Spd-2-ALL- or Spd-2-CONS-fusion protein function the control is always the WT Spd-2-fusion tested at the same gene dosage; (2) The Spd-2-11A mutant protein is expressed at much lower levels than the WT protein (Figure 1—figure supplement 1A), yet it localises to centrosomes nearly as strongly as the WT protein (Figure 1—figure supplement 1B-D). Thus, the reduction in the centrosomal levels of Spd-2-ALL and Spd-2-CONS is unlikely to simply be due a general reduction in their protein levels. We now discuss this point.

4) The images of centrosomes in Figures 4-6 appear to be S-phase centrosomes with flares, not mitotic centrosomes. The authors reference mitosis and maturation when, in fact, they show images of interphase (S-phase) centrosomes. The authors should show low magnification images of mitotic spindles and then high magnification insets of centrosomes to demonstrate that they are, in fact, examining mitotic centrosomes.

The reviewer questions whether we are examining centrosomes in interphase or mitosis. We apologise for not explaining this properly. In these rapidly developing embryos the nuclei progress through successive S- and M-phases without intervening Gap-phases. As a result, the centrosomes are essentially always either *in* mitosis, or are *preparing to enter* mitosis—they are never truly in an “interphase” state where they organise only the tiny amounts of PCM one might observe in a fly somatic cell in interphase. Thus, as soon as mitosis finishes the embryos enter S-phase and the centrosomes start to mature in preparation for the next round of mitosis. We now clarify this important point.

Reviewer #3:[…] 1) The authors should provide direct evidence about how much Spd-2-CONS and Spd-2-ALL are affects their binding with the Polo PBD compared to WT, which is critical for most of their conclusions. I'm concerned that mutating all 34 Serine sites may affect protein structure and this needs to be ruled out.

Please see point 1 of our response to reviewer #1: we now show that a recombinant WT Spd-2 MBP-fusion protein can bind to the PBD when phosphorylated by Plk1, but mutation of the S-S/T motifs to T-S/T prevents binding (New Figure 1C). The reviewer was also concerned that the multiple Ser-The substitutions we introduce in the mutant proteins may affect protein structure. Please see point 3 in our response to reviewer #2: we now present several lines of evidence that indicate that the mutant proteins are not simply misfolded.

2) Do the authors now if the phosphorylation status is altered during the cell cycle?

This is a good question that we have tried to address by raising phospho-specific antibodies to Spd-2—but so far without success.

3) Figure 4: authors conclude that the inability of mutant proteins to recruit Polo reduces the assembly and/or maintenance of this mixed scaffold, but the data shown can't exclude the possibility that mutant proteins are efficiently incorporated into the scaffold. The author should show a comparison of endogenous Spd-2 scaffold in all three conditions.

We apologise that we were unclear on this, as we do indeed believe that the mutant Spd-2 proteins can incorporate into the scaffold formed by the WT protein. This is entirely consistent with our model: as long as there is *some* WT Spd-2 in the scaffold, it should recruit Polo, which can then phosphorylate Cnn to support the expanding Spd-2 scaffold; thus, there is no requirement that *every* Spd-2 molecule in the scaffold be able to recruit Polo. As requested by the reviewer, we now directly address this issue by expressing either the WT or mutant Spd-2-GFP fusions in the presence of WT Spd-2-mCherry (but in the absence of any endogenous unlabelled Spd-2). This allows us to directly compare the ability of the WT and mutant GFP-fusions to incorporate into the expanding scaffold formed by the WT Spd-2-mCherry (New Figure 5). The incorporation of the WT and mutant Spd-2-GFP fusion proteins into the WT Spd-2-mCherry scaffold is essentially indistinguishable—consistent with our model.

At a first glance, this new data may appear at odds with the data shown in Figure 4B of our original submission (now superseded by New Figure 5). In this earlier experiment we showed that in the presence of WT unlabelled Spd-2, the scaffold formed by the mutant-GFP fusion proteins appears somewhat weaker than that formed by the WT-GFP fusion protein—suggesting that the presence of the mutant protein may partially inhibit the assembly of the WT scaffold (which is also consistent with our model, as presumably less Polo will be recruited to the PCM in the presence of the mutant proteins). These two experiments, however, are not comparable: in the old experiment, the scaffold is formed from 2 copies of the endogenous unlabelled *Spd-2* gene and two copies of the *Spd-2-GFP* fusion (Old Figure 4B) while in the new experiment it is formed from one copy of *WT-Spd-2-mCherry* and one copy of the *Spd-2-GFP* fusions in the absence of any endogenous (unlabelled) protein (new Figure 5). Thus, there are several possible explanations for this minor discrepancy (for example, unlabelled Spd-2 might more efficiently compete with mutant Spd-2-GFP for incorporation into the scaffold than WT-Spd-2-mCherry, so the ratio of WT/mutant proteins in the scaffolds may be very different).

4) Figure 5: I'm not convinced by the authors' conclusion that the pericentriolar scaffold cannot expand because Spd-2-CON can't recruit Polo. First, they co-localize at the mother centriole, and second, Spd-2-CON itself is affecting scaffold formation in Spd-2 mutant background (Figure 4), so how can they be certain it's a Polo recruitment issue? I would potentially suggest a FRAP experiment to showcase the recruitment of Polo to the scaffold. In addition, apart from co-expressing with Polo-GFP, what's the difference between Figure 5B and Figure 4A (middle lane)? Why is there's a huge difference between the ratios (64/36 vs 9/91)? This was confusing to me.

The reviewer raises two specific reasons for this scepticism. First, the mutant proteins still co-localise with Polo at the centriole (suggesting that they can still recruit Polo). Please see point 2 in our response to reviewer #2where we discuss why we think that Spd-2 is not required to recruit Polo to centrioles. Second, although the Spd-2 mutants effect scaffold assembly we cannot be certain that this is due to a Polo recruitment defect. Please see point 2 of our response to reviewer #1where we describe new experiments that address this problem.

The reviewer suggests that we use FRAP to measure the dynamics of Polo recruitment to the scaffold. This is a good experiment, but we believe it would be hard to interpret. This is because Polo may have different turnover rates at the centriole and in the PCM and the ratio of Polo in these two structures is very different in embryos expressing WT Spd-2-mCherry (where it is mostly in the PCM) and mutant Spd-2-mCherry (where it is mostly in the centriole).

The reviewer asks why the PCM recruitment of the mutant Spd-2-CONS protein is so much worse in embryos expressing Polo-GFP (New Figure 6B) than in embryos not expressing Polo-GFP (New Figure 4). We apologise for not explaining this more clearly. The Polo-GFP line we use here is the healthiest available (an exon-trap insertion into the endogenous *Polo* gene), but the GFP tag partially disrupts Polo function and this line is only viable in the presence of an untagged copy of *Polo*. As we now explain more fully, there appears to be a strong genetic interaction between *mutant* Spd-2 proteins and the partially functional Polo-GFP. Embryos lacking endogenous Spd-2 and expressing Polo-GFP are viable in the presence of WT Spd-2-mCherry, but die very early in the presence of Spd-2-ALL-mCherry or Spd-2-CONS-mCherry (and much earlier than embryos expressing just Spd-2-CONS-mCherry or Spd-2-ALL-mCherry without the Polo-GFP). We do not understand the basis for this genetic interaction, but it suggests an intimate link between Spd-2 and Polo that is somehow perturbed when Spd-2 is mutated in this way *and* Polo is GFP-tagged.

5) Figure 7: what is the component of the scaffold? If the scaffold is made of Spd-2, when Spd-2 CONS is unable to form the scaffold, you won't be able to see other proteins there no matter how they are recruited. Is there any other marker for the scaffold?

This is correct. However, although we have long proposed that Spd-2, Polo and Cnn cooperate to form the PCM scaffold in flies, we feel that this is not yet an established fact. Hence, we sought to test whether other PCM components might still be recruited to an expanded PCM that might form independently of Spd-2 and Polo. Our data suggest that this is not the case, supporting the view that Spd-2, Polo and Cnn are the major components of the PCM scaffold. The reviewer asks if there is any other marker of the scaffold we can test, but these are the only three markers that we are aware of.

6) Figure 9: The C-terminus of Spd-2 contains most of the conserved Serine sites, but the authors show that mutation N-terminal sites have the biggest effect? What do the authors think the molecular basis for this is?

We apologise for being unclear on this point. The N- and C-terminal halves of Spd-2 that we test here each have 17 S-S/T motifs (see Figure 2A) and we mutated all 17 of these (not just the conserved ones) in the experiments comparing the effect of the mutations in each half of the protein (Figure 9). We have now clarified this point. We do not know why the N-terminal mutations have a larger effect, but suspect that the N-terminal half of Spd-2 normally plays the greater part in recruiting Polo to the PCM in vivo.

[Editors' note: the author responses to the re-review follow.]

Reviewer #1:[…] Regarding the immunoprecipitation (IP) experiments between the MBP-phospho-Spd2 fragment and GST-PBD (new Figure 1C), the resolution of these immunoblots appear low and the brightness/contrast altered such that it is difficult to gauge the load between the samples. It appears that there is more GST-PBD in the MBP-Spd-2 WT immunoprecipitate, but how much more? Replacing these with higher resolution blots with accompanying quantitation would be appropriate. This is an important experiment/result because it demonstrates that the authors have in fact generated a Spd-2 Polo-binding mutant. Also, is the kinase being used human Plk1, whereas the Spd-2 and PBD proteins are fly? Whatever the species, the fact that the authors used proteins from different organisms should be stated in the Results section.I also recommend a couple of text changes in this section. The authors state that binding of the 19T mutant was reduced to background levels. How do they know this is background? This could be real binding and, therefore, would remain at WT levels, not a reduced level. The authors end this section stating, "Thus, a fragment of Spd-2 can bind directly to the PBD when phosphorylated, and this binding is prevented when the S-S/T motifs are mutated to T-S/T". Based on the data, it looks to me that binding is not prevented. Instead, they should say that phosphorylation of Spd-2 increases PDB binding which, in my opinion, is what the data shows. The data also suggest that Polo activity primes it's binding to Spd-2 (as they state in the Discussion) but this important finding should also be emphasized here at the end of this Results section.

The reviewer made a number of points about our presentation and description of the immunoprecipitation experiment showing that phosphorylated Spd-2 fragment can bind recombinant GST-PBD in vitro (Figure 1C). These points were all valid and we have made the following changes:

1) We now show the quantification from three repeats of the experiment and state in the Results section that we use recombinant human Plk1 and a fragment of *Drosophila* Spd-2 in these experiments.

2) We agree that our original description of this new experiment was unclear in several respects. We have now rewritten this section to focus on the conclusion that the MBP-Spd-2 fragment binds more GST-PBD when phosphorylated by Plk1, and that this increase in binding is not seen when the S-S/T sites have been mutated to T-S/T sites.

3) We now emphasise the important finding that Polo activity may prime its own binding to Spd-2 in both the Results and Discussion sections.

Reviewer #2:

*[…] The first concern was dealt with by providing* in vitro *biochemical interaction data (Figure 1C) and the second by performing a new type of quantification of the centrosomal Polo signal in cells expressing a mix of WT and mutant Spd-2, at a time point when Polo signal is just starting to become reduced (Figure 10). This approach allowed the authors to demonstrate reduced Polo signal even in cells where a Spd-2 PCM scaffold is still present. Although the data supports the authors' conclusion that Spd-2 mutants cause a specific loss of PCM associated Polo, the quantification would be more convincing if actual intensities would have been quantified for the Spd-2 scaffold and Polo signals rather than subjective classification into "weak" and "strong". This is because in contrast to a similar quantification in Figure 6, where an "all or nothing" effect is observed, here the relative intensities of PCM scaffold-associated Spd-2 vs Polo are crucial for the result. I would suggest to also adding quantification of intensities to the figure.*

My other concerns were also addressed, not in all cases with experiments, but I recognize the technical challenges that have led to this decision.I would now support publication.

The reviewer requested that we provide some additional quantification of fluorescent intensities for the experiment shown in Figure 10, rather than the more qualitative classification of images into “weak” or “strong” PCM signal. The reason we do not do this (for this experiment and for several others where we use 3D-SIM to assess the localisation of proteins at both the centriole and in the PCM) is that these 3D-SIM images should not really be used for intensity quantification as they are computer-based reconstructions of what the computer “thinks” the original image looks like. Our 3D-SIM experts in the Micron Oxford Advanced Bioimaging Facility tell us that this makes it unreliable to compare absolute intensity levels from such images, particularly if they have been collected from different samples. This is why in all figures like this we show several representative images to cover the range of phenotypes we observe and quantify in our classification of the reconstructed images. As all scoring is performed blind, we are confident that this “semi-quantitative” scoring system provides a reproducible and accurate reflection of the underlying data.

In addition we have made a few minor textual clarifications, corrected a few typos, and inserted an extra reference to a very recently published paper by Cabral et al., 2019, that is relevant to this work.

References:

1. Lee, K. S., Grenfell, T. Z., Yarm, F. R. & Erikson, R. L. Mutation of the polo-box disrupts localization and mitotic functions of the mammalian polo kinase Plk. *Proceedings of the National Academy of Sciences* 95, 9301–9306 (1998).

2. Song, S., Grenfell, T. Z., Garfield, S., Erikson, R. L. & Lee, K. S. Essential function of the polo box of Cdc5 in subcellular localization and induction of cytokinetic structures. *Mol Cell Biol* 20, 286–298 (2000).

3. Seong, Y.-S. et al. A spindle checkpoint arrest and a cytokinesis failure by the dominant-negative polo-box domain of Plk1 in U-2 OS cells. *J Biol Chem* 277, 32282–32293 (2002).

4. Reynolds, N. & Ohkura, H. Polo boxes form a single functional domain that mediates interactions with multiple proteins in fission yeast polo kinase. *J Cell Sci* 116, 1377–1387 (2003).

5. Elia, A. E. H., Cantley, L. C. & Yaffe, M. B. Proteomic screen finds pSer/pThr-binding domain localizing Plk1 to mitotic substrates. *Science* 299, 1228–1231 (2003).

6. Elia, A. E. H. et al. The molecular basis for phosphodependent substrate targeting and regulation of Plks by the Polo-box domain. *Cell* 115, 83–95 (2003).

7. Liu, J., Lewellyn, A. L., Chen, L. G. & Maller, J. L. The polo box is required for multiple functions of Plx1 in mitosis. *J Biol Chem* 279, 21367–21373 (2004).

8. Decker, M. et al. Limiting amounts of centrosome material set centrosome size in C. elegans embryos. *Curr. Biol* 21, 1259–1267 (2011).

9. Joukov, V., De Nicolo, A., Rodriguez, A., Walter, J. C. & Livingston, D. M. Centrosomal protein of 192 kDa (Cep192) promotes centrosome-driven spindle assembly by engaging in organelle-specific Aurora A activation. *Proc Natl Acad Sci USA* 107, 21022–21027 (2010).

10. Joukov, V., Walter, J. C. & De Nicolo, A. The Cep192-organized aurora A-Plk1 cascade is essential for centrosome cycle and bipolar spindle assembly. *Mol Cell* 55, 578–591 (2014).

11. Meng, L. et al. Bimodal Interaction of Mammalian Polo-Like Kinase 1 and a Centrosomal Scaffold, Cep192, in the Regulation of Bipolar Spindle Formation. *Mol Cell Biol* 35, 2626–2640 (2015).

12. Novak, Z. A., Wainman, A., Gartenmann, L. & Raff, J. W. Cdk1 Phosphorylates Drosophila Sas-4 to Recruit Polo to Daughter Centrioles and Convert Them to Centrosomes. *Dev Cell* 37, 545–557 (2016).

13. Wueseke, O. et al. The Caenorhabditis elegans pericentriolar material components SPD-2 and SPD-5 are monomeric in the cytoplasm before incorporation into the PCM matrix. *Mol Biol Cell* 25, 2984–2992 (2014).